



# Fine scale zooplankton distribution across the North Balearic Front during late spring

Maxime Duranson[1,2], Léo Berline[1], Loïc Guilloux[1,3], Alice Della Penna[4,5], Mark D. Ohman[6], Sven Gastauer[2], Cédric Cotte[7], Daniela Bănaru[1], Théo Garcia[8], Maristella Berta[9], Andrea Doglioli[1], Gérald Gregori[1], Francesco D'Ovidio[7], and François Carlotti[1]

[1]Université Aix Marseille, Université de Toulon, CNRS, IRD, MIO, Marseille, France
[2]Thünen Institute of Sea Fisheries, Bremerhaven, Germany
[3]CNRS,Univ Brest, IRD, IFREMER, LEMAR, IUEM, F-29280 Plouzané,France
[4]Institute of Marine Science, University of Auckland, New Zealand
[5]School of Biological Sciences, University of Auckland, New Zealand
[6]Integrative Oceanography Division, Scripps Institution of Oceanography, La Jolla, California, USA
[7]Sorbonne Université, CNRS, IRD, MNHN, Laboratoire d'Océanographie et du Climat: Expérimentations et Approches Numériques (LOCEAN-IPSL), Paris, France.
[8]Aix Marseille University CNRS, Centrale Marseille, I2M, Marseille, France
[9]Istituto di Scienze Marine- Consiglio Nazionale delle Ricerche (ISMAR-CNR), Sede Secondaria di Lerici, 19032 La Spezia, Italy

*Correspondence to:* Maxime Duranson (maxime.duranson@thuenen.de), François Carlotti (francois.carlotti@mio.osupytheas.fr)

## Keywords

BioSWOT-Med campaign, Northwestern Mediterranean Sea, Ocean Fronts, Vertical distributions, Copepod Community, Size, Trophic categories

## Abstract

Observations, models and theory have suggested that ocean fronts are ecological hotspots, generally associated with higher diversity and biomass across many trophic levels. Nutrient injections are often associated with higher chlorophyll concentrations at fronts, but the response of the zooplankton community is largely unknown. The present study investigates mesozooplankton stocks and composition during late spring, northeast of Menorca along two north-south transects that crossed the North Balearic Front (NBF) separating central water of the Northwestern Mediterranean Sea (NWMS) gyre from peripheral waters originating from the Algerian basin. During the BioSWOT-Med campaign, samples were collected using vertical triple net tows at three depths (100 m, 200 m, 400 m) with 200 $\mu$m and 500 $\mu$m mesh nets, processed with ZooScan, and the organisms assigned to eight taxonomic groups. Zooplankton distributions were analysed for the surface (0–100 m), intermediate (100–200 m), and deeper (200–400 m) layers. The results showed no significant biomass increase at the front across all vertical layers. The NBF seems to act more like a boundary between communities rather than a pronounced area of active or passive zooplankton accumulation. Analysis of stratified vertical distributions of zooplankton highlighted distinct taxonomic compositions in the





surface, intermediate, and deeper layers, and a progressive homogenization of community structure with depth, reflecting a weaker impact of hydrological processes on deeper communities. The front's clearest impact was within the upper 100 meters,

where the taxonomic composition showed differences between the front and the adjacent water masses, with a decrease in Eumalacostraca and Foraminifera, while Cnidaria increased sharply. In the 100–200 m layer, the front also influenced community composition, although to a lesser extent, with a marked increase in Foraminifera and a strong decline in Cnidaria. Moreover, the northern water mass and the front were dominated by large copepods, while the southern water mass exhibited higher diversity of zooplankton and a community of smaller-sized copepods. The results of this study highlight the complexity of pro-

cesses shaping planktonic communities over time and space in the NBF zone and its adjacent waters. These processes include zooplankton stock reduction in the transitional post-bloom period, marked effect of diel variation linked to vertical migrations, and potentially the impact of storm-related mixing in the surface layer that can disrupt established ecological patterns.

# 1 Introduction

Fronts are key structural features of the ocean, affecting all trophic levels across a wide range of spatial and temporal scales

(Belkin et al. (2009)). Oceanic fronts are narrow regions of elevated physical gradients that separate water parcels with distinct properties, such as temperature, salinity, and thus density (Hoskins (1982); Joyce (1983); Pollard and Regier (1992); Belkin and Helber (2015)). These frontal zones act as dynamic boundaries between distinct water masses (Ohman et al. (2012); Mańko et al. (2022)), which play a crucial role in shaping marine ecosystems (Belkin et al. (2009)). Moreover, fronts display wide variations in spatial and temporal dimensions ranging from hundreds of meters to tens of kilometres, and from short-lived to

permanent (Owen (1981); McWilliams (2016); Lévy et al. (2018)).

The relationships between fronts and plankton have received considerable attention in marine ecology due to the enhanced biological production and community changes that are sometimes observed in their vicinity (Le Fèvre (1987); Fernández et al. (1993); Pinca and Dallot (1995); Errhif et al. (1997); Pakhomov and Froneman (2000); Chiba et al. (2001); Munk et al. (2003)). As physical barriers or zones of mixing, fronts structure biomass and species distributions, generally leading to distinct

ecological communities on either side (Ohman et al. (2012); Le Fèvre (1987); Prieur and Sournia (1994); Gastauer and Ohman (2024)). Fronts are also known to concentrate high phytoplankton concentration and, as a result, higher biomass levels and metabolism of zooplankton (Thibault et al. (1994); Ashjian et al. (2001); Ohman et al. (2012); Derisio et al. (2014); Powell and Ohman (2015a)). Frontal structures can enhance primary and secondary production essentially by promoting nutrient input through cross-frontal mixing and vertical circulation driven by horizontal density gradients (Durski and Allen (2005); Liu

et al. (2003); Derisio et al. (2014); Russell et al. (1999). These nutrient-rich conditions (bottom-up) sometimes fuel elevated chlorophyll concentrations, supporting the aggregation of zooplankton, fish larvae, and their higher trophic level predators such as tuna, sharks, seabirds and whales (Herron et al., 1989Herron et al. (1989); Olson et al. (1994); Royer et al. (2004); Queiroz et al. (2012); di Sciara et al. (2016); Druon et al. (2019)). Pronounced changes in zooplankton Diel Vertical Migration behaviour have also been observed across frontal gradients (Powell and Ohman (2015b), Gastauer and Ohman (2024)).





Recent studies (Mangolte et al. (2023); Panaïotis et al. (2024)) have highlighted the importance of investigating zooplankton at fine scales and their patchiness in the vicinity of fronts to understand their interactions with particles and the environment. Mangolte et al. (2023) revealed that the plankton community exhibits fine-scale variability across fronts, with biomass peaks of different taxa often occurring on opposite sides of the front, or with varying widths. This fine-scale cross-frontal patchiness suggests processes that spatially decouple plankton taxa, leading to the formation of multiple adjacent communities rather than

a single coherent frontal plankton community.

In the Northwestern Mediterranean Sea (NWMS), the role of mesoscale structures in the open ocean such as density fronts and eddies on the distribution and diversity of zooplankton has already been widely documented (Saiz et al. (2014)). These structures generally increase the patchiness and activity of plankton, and stimulate trophic transfers to large predators (Cotté et al. (2009), Cotté et al. (2011)). Due to its coastal proximity, the North Current's frontal zones have been widely studied from physical and ecosystem perspectives, on both the Ligurian (Prieur et al. (1983); Stemmann et al. (2008)) and Catalan

sides (Font et al. (1988); Sabatés et al. (2007)). Downstream of the Northern Current, the North Balearic current flows from northeast Menorca to southwest Corsica. This current is associated with the North Balearic Front (NBF), which marks the transition zone between the saltier surface waters from the Provencal basin at its northern edge, and the less salty surface waters from the Algerian basin at its southern edge. To the south is the Atlantic water that has recently penetrated the Strait of

Gibraltar, while to the north is the Atlantic water that has circulated in the western basin and has consequently become saltier due to evaporation typical of the Mediterranean Sea (Millot and Taupier-Letage (2005)). Recent contributions from glider data and satellite imagery have enabled us to better characterize the NBF (Barral (2022)). Its latitudinal position shows seasonal (from 40.2°N in spring to 41°N in autumn) and interannual variations, notably linked to the intensity and extension of the winter deep convection in the Provençal basin to the north, and to the south, where, mainly in summertime, mesoscale dynamics

spread lighter Atlantic water northward between Menorca and Sardinia (Millot (1999); Seyfried et al. (2019)). Among the most pronounced geostrophic frontal zones in the NWMS, the NBF and its ecological impacts are the least studied.

The BioSWOT-Med cruise, focused on the NBF, involved a wide range of sampling and data collection, including zooplankton sampling using nets. In this study, the following questions were addressed: how are zooplankton communities structured on each side of the front; is the zooplankton community at the front a mixture of communities from both sides, or does it form

a distinct community; does the front influence the vertical structure of zooplankton communities; and can weather events, such as storms, influence the structure of zooplankton communities within the water masses?

## 2   Materials and methods

### 2.1   Study area and sample strategy

The BioSWOT-Med cruise (https://doi.org/10.13155/100060; PIs: A. Doglioli and G. Grégori) was performed on board the

R/V L'Atalante (FOF-French Oceanographic Fleet) from 21 April to 14 May 2023 in an area about 100 km north-east of




Menorca Island (NWMS) (Fig. 1). Figure 1.a shows the zone as observed four days before the first transect due to cloud cover during this part of the survey (Fig. A1).

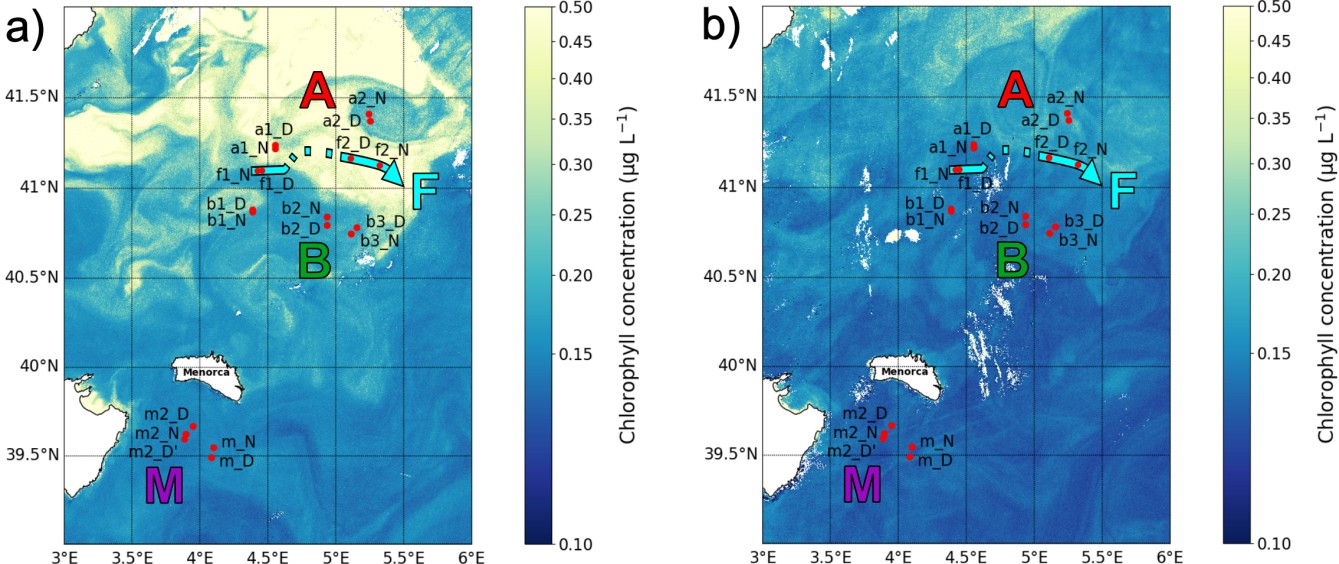

**Figure 1.** Maps of the sampling stations with surface chlorophyll concentration ($\mu$g L-1) from Sentinel 3. a) Map from April 21 showing conditions 4 days before the first transect. b) Map from May 5 showing conditions during the second transect. The colours representing the three water masses and the front will be maintained throughout the paper.

The strategy of the cruise was designed to take advantage of the novel SWOT satellite mission, in order to better resolve fine-scale oceanic features. During the "fast sampling phase," SWOT provided altimetry data characterized by high spatial resolution and a 1-day revisit period over 150 km-wide oceanic regions. With the support of the international SWOT AdAC (Adopt-A-Crossover, https://www.swot-adac.org/; PI F.d'Ovidio) Consortium, the BioSWOT cruise applied an adaptive multidisciplinary approach by combining daily SWOT images and environmental bulletins provided daily by the SPASSO toolbox (https://spasso.mio.osupytheas.fr/, Rousselet et al., submitted), and in situ measurements using a suite of instruments to capture physical, biological, and chemical properties (Doglioli et al. (2024), cruise report). This strategy enabled the targeting of fine-scale features of the North-Balearic front.

This zone separates two water masses: the northern water mass, referred to as "A", which is more productive, colder, and characterised by higher salinity, and the southern mass, referred to as "B", which is less productive, warmer, and less saline (Fig. 1). This permitted us to sample two contrasting water masses separated by a sharp frontal region, designated "F". The sampling strategy focused on two main transects, covering a total of six 24-hour drifting stations: A-F-B (to the west) and B-F-A (to the east), in addition to three supplementary stations: *b3*, *m*, and *m2* (Fig. 1 and Table 1). The two f2 stations are relatively distant from each other due to a strong frontal current. Because of a storm (2nd May), a third area, "M", was sampled twice while the ship took shelter south of Menorca and conducted similar measurements as in zones A, B, and F. The M zone



is different from the three other sampled zones in terms of bathymetry, as it is located around 20 km from the continental shelf. On the way home, a final station was sampled in B (Table 1). At every station, physical properties were sampled using a CTD rosette, which was deployed four times daily at fixed intervals (06:00, 12:00, 18:00, and 00:00 local time). Hereafter, water masses will be designated by an uppercase letter (A, B, M, and F for the front), and stations by a lowercase letter ($a$, $b$, $m$, $f$).

**Table 1.** Stations details. In Abbreviation, 'D' stands for Day and 'N' stands for Night.

| Campaign Stage | Station Name | Abbreviation | Date - Time | Latitude | Longitude |
|---|---|---|---|---|---|
| 1st transect | a1_zoo1 | a1_D | 25/04 - 12:38 | 41.240 | 4.553 |
| | a1_zoo2 | a1_N | 26/04 - 00:02 | 41.224 | 4.563 |
| | f1_zoo1 | f1_N | 27/04 - 00:32 | 41.099 | 4.423 |
| | f1_zoo2 | f1_D | 27/04 - 12:11 | 41.102 | 4.456 |
| | b1_zoo1 | b1_N | 28/04 - 00:17 | 40.874 | 4.388 |
| | b1_zoo2 | b1_D | 28/04 - 12:28 | 40.884 | 4.389 |
| Storm | m_zoo1 | m_N | 02/05 - 00:37 | 39.555 | 4.101 |
| | m_zoo2 | m_D | 02/05 - 12:22 | 39.493 | 4.087 |
| 2nd transect | b2_zoo1 | b2_D | 04/05 - 12:16 | 40.795 | 4.933 |
| | b2_zoo2 | b2_N | 05/05 - 00:13 | 40.849 | 4.936 |
| | f2_zoo1 | f2_D | 05/05 - 11:49 | 41.175 | 5.108 |
| | f2_zoo2 | f2_N | 06/05 - 00:45 | 41.134 | 5.308 |
| | a2_zoo1 | a2_N | 07/05 - 00:13 | 41.412 | 5.240 |
| | a2_zoo2 | a2_D | 07/05 - 12:15 | 41.376 | 5.253 |
| Return water mass M | m2_zoo1 | m2_D | 10/05 - 11:31 | 39.671 | 3.957 |
| | m2_zoo2 | m2_N | 11/05 - 00:31 | 39.629 | 3.902 |
| | m2_zoo3 | m2_D' | 11/05 - 11:53 | 39.603 | 3.885 |
| Return water mass B | b3_zoo1 | b3_D | 12/05 - 12:15 | 40.782 | 5.152 |
| | b3_zoo2 | b3_N | 12/05 - 23:58 | 40.746 | 5.112 |

## 2.2 Zooplankton collection

Zooplankton samples were collected using a triple net (Triple-WP2) equipped with three individual nets, each with a 60 cm mouth diameter but different mesh sizes (500 $\mu$m, 200 $\mu$m, 64 $\mu$m). For this study, which focuses on mesozooplankton, only the samples collected by 200 and 500 $\mu$m nets were used. The nets were deployed vertically to cover three integrated layers (400-0 m; 200-0 m; 100-0 m). Sampling occurred twice daily at local times: 12:00 and 00:00. Note that the net deployed to 400 m at station $m\_N$ (Table 1) could not be analysed because it was found folded up on itself upon retrieval. After collection, samples were preserved in 4% borate-buffered formaldehyde.

## 2.3 Zooplankton sample processing

In a shore-based laboratory, samples were digitized with the ZooScan digital imaging system (Gorsky et al. (2010)) to identify and determine the size structure of the zooplankton communities. Each sample was divided into one of two size fractions ($<1000$ and $>1000$ $\mu$m) for better representation of rare large organisms in the scanned subsample (Vandromme et al. (2012)). Each fraction was split using a Motoda box (Motoda (1959)) until it contained an appropriate number of objects, approximately 1500. After scanning, each image was processed using ZooProcess (Gorsky et al. (2010)), which is written



in the ImageJ image analysis software (Rasband (1997–2011)). Only objects having an Equivalent Circular Diameter (ECD) > 300 $\mu$m were detected and processed (Gorsky et al. (2010)). Objects were first automatically classified using ECOTAXA (https://ecotaxa.obs-vlfr.fr/), then the results were validated by a taxonomist. With this approach 101 taxa were detected which were then grouped in 8 categories: Appendicularia, Chaetognatha, Copepoda, Cnidaria, Eumalacostraca, Foraminifera, Thaliacea and Other_Organisms (Table 2). This last category includes all taxa which would not belong in any of the designated classes and which were present in very low numbers across all samples. Zooplankton abundance (number of individuals $m^{-3}$) was calculated from the number of validated vignettes in ZooScan samples, considering the scanned fraction and the sampled volume from the nets.

We also took into account trophic differences between the 40 taxonomic groups of copepods by assigning them into five major trophic groups based on their feeding behaviour, i.e., filter feeders, carnivorous cruise feeders, carnivorous ambush feeders, non-carnivorous cruise feeders, and non-carnivorous ambush feeders, following the classification by Benedetti et al. (2015) (Table 2). Non-carnivorous copepods include herbivores and detritivores, and filter feeders include mixed feeders. The difference between cruise and ambush feeders is that cruise feeders have to swim actively to encounter their prey, while ambush feeders passively encounter them (Kiørboe et al. (2015)). The taxonomic categories other than copepods have well identified trophic behaviour, and were grouped in the different trophic groups (Table 2).

**Table 2.** Assignments of trophic groups to identified taxonomic classifications.

| Trophic groups | Categories | Abbreviations | Taxonomic Groups identified by ZooScan |
|---|---|---|---|
| Carnivores | Chaetognatha | Cha | Chaetognatha |
| | Copepods Carnivores Ambush-Feeders | Cop CAF | *Candacia*, Corycaeidae (*Corycaeus, Urocorycaeus*), *Lubbockia* |
| | Copepods Carnivores Cruise-Feeders | Cop CCF | Aetideidae*, Euchaeta, Haloptilus, Heterorhabdus*, Sapphirinidae (*Copilia, Sapphirina, Vettoria*), *Urocorycaeus* |
| | Cnidaria | Cni | *Cnidaria (ephyra), Trachylinae (Aglaura, Solmundella), Obelia, Siphonophorae, Calycophorae (Diphyidae, Chelophyes, Chelophyes appendiculata, Eudoxoides, Eudoxoides spiralis, Hippopodius, Hippopodius hippopus, Lensia, Lensia subtilis, Muggiaea)*, Physonectae |
| | Foraminifera | For | Foraminifera |
| Filter-feeders | Appendicularians | App | Fritillariidae, Oikopleuridae |
| | Copepods Filter-Feeders | Cop FF | *Acartia, Calanus helgolandicus, Calocalanus, Centropages, Centropages bradyi, Centropages typicus, Eucalanus, Lucicutia, Lucicutia ovalis, Mormonilla, Nannocalanus minor, Pleuromamma, Pleuromamma abdominalis, Rhincalanus, Subeucalanus, Temora* |
| | Thaliacea | Tha | Doliolida, Pyrosomatida, Salpida (*Salpa fusiformis*) |
| Herbivores & detritivores | Copepods Non-Carnivores Ambush-Feeders | Cop NCAF | *Distioculus minor, Oithona* |
| | Copepods Non-Carnivores Cruise-Feeders | Cop NCCF | *Arietellus, Euchirella messinensis, Euchirella rostrata, Euterpina acutifrons, Mecynocera clausi, Microsetella*, Oncaeidae, Peltidiidae, Scolecitrichidae |
| Omnivores | Eumalacostraca | Eum | Amphipoda (Hyperiidae undetermined, *Phronima, Phrosina, Platyscelus, Primno, Pronoe, Scina, Vibilia*), Decapoda (Dendrobranchiata, Penaeoidea), Euphausiacea larvae, Isopoda, Mysida |
| All | Other_Organisms | Oth | Bivalvia larvae, Bryozoa larvae, Cirripedia larvae, Crustacea nauplii, Ctenophora, Cyphonaute, Echinodermata larvae, *Evadne*, Teleostei (larvae and eggs), Gastropoda (Atlantidae, Cavoliniidae, Cliidae, Creseidae, Cymbulioidea, Limacinidae), Halosphaera, Heteronemertea, Ostracoda, Polychaeta larva (Lopadorrhynchidae, Magelonidae, Phyllodocidae, Tomopteridae, Typhloscolecidae), Radiolaria, Tintinnida |





Enumerations from the 200 $\mu$m and 500 $\mu$m nets, which were deployed simultaneously, were combined. To avoid double counting of organisms large enough to be captured by both nets, a threshold value was established, based on the analysis of the Normalized Biomass Size Spectra (NBSS) (Sect. 2.7), considering all stations and depths (a specific value for each station would not have significantly altered the results). The threshold value (1148 $\mu$m ECD) identified the body size at which the 500 $\mu$m net samples more effectively (Fig. A2). Thus, organisms smaller than this size from the 200 $\mu$m net, and those larger from

the 500 $\mu$m net, were combined to form a new count, called 'combined net' hereinafter.

## 2.4    Calculation of biovolumes

The biovolume of each organism can be calculated in different ways, including from ECD (Heath (1995)), or by determining the ellipsoid that best fits the object. However, mesozooplankton are generally not spherical in shape, meaning the ECD approach tends to overestimate biovolume because spheres have a higher volume-to-cross-sectional area ratio compared to other shapes

(Sprules et al. (2003); Beaulieu et al. (1999)). Consequently, a more accurate approach involves calculating biovolume based on the ellipsoid that best fits the organism (Alcaraz et al. (2003)). However, since the ellipsoid is a 3D object and the ZooScan image provides only 2D data, the following approach is used: (i) an ellipse is fitted to the 2D object, defining the major and minor axes; (ii) in 3D, the two minor axes are assumed to be equal; (iii) the biovolume is then calculated using formula (1) (Dubois et al. (2021)):

$$B = \frac{4}{3} \times \pi \times \text{major} \times \text{minor}^2 \tag{1}$$

## 2.5    Definition of intermediate layers: 100-200 m and 200-400 m

Our nets sampled the layers: 0-100, 0-200, and 0-400 meters (Sect. 2.2). In order to study the community as a function of depth, the abundance of different taxonomic groups (Sect. 2.3) was calculated in each layer by differencing. For instance, subtracting the abundance measured at 0-100 m from that at 0-200 m provided the abundance of each taxonomic group, trophic group,

or size class for the layer 100-200 m. A similar approach was applied to determine the abundance of taxonomic groups in the 200-400 m layer. This approach is assumed valid as the net's tows were carried out successively within a relatively short interval of time, typically 45 minutes. It is important to note that subtractions were performed on the eight major categories and not on each taxonomic group (see Table 2). In rare cases (12 %), especially for Eumalacostraca (particularly in the 100-200 m layer) and Cnidaria (particularly in the 200-400 m layer), resulting abundances were negative and thus set to zero.

## 2.6    Analysis of variance and Post-Hoc comparisons

Using R version 4.4.1 (Team (2025)), one-way analyses of variance (ANOVA) was conducted to examine differences in absolute abundance across groups for each taxonomic category. Prior to performing the ANOVA, the normality of the data distribution was assessed using the Shapiro-Wilk test and the homogeneity of variances verified with Levene's test (car package, version 3.1-3; Fox and Weisberg (2019)). ANOVAs were then performed for five factors: Water masses, Layer, Period (day or





night), Transects (storm effect) and Copepod subgroups (mean abundances of the eight most abundant taxa within copepods). For each significant ANOVA result ($p < 0.05$), a Tukey's Honest Significant Difference test was applied to identify the groups that differ substantially from one another.

## 2.7 Normalized Biomass Size Spectra (NBSS)

The size of organisms is considered a key indicator of community dynamics (Platt and Denman (1977)). NBSS (Platt and
Denman (1977)) are widely used to study this property. For constructing the NBSS, zooplankton organisms are grouped into logarithmically increasing size classes. The total biovolume of each class is then divided by the width of its size class (Platt and Denman (1977)). The x-axis [log2 zooplankton biovolume (mm$^3$/ind)] is calculated as:

$$\log_2 \left[ \frac{\text{Zooplankton biovolume (mm}^3.\text{m}^{-3})}{\text{Abundance of each class size (ind.m}^{-3})} \right] \tag{2}$$

The y-axis [log2 normalized biovolume (m$^3$)] is calculated as:

$$\log_2 \left[ \frac{\text{Zooplankton biovolume (mm}^3.\text{m}^{-3})}{\text{Interval of each class size } (\Delta\text{volume.mm}^{-3})} \right] \tag{3}$$

The NBSS thus represents the normalized biovolume as a function of the size of the organisms, both on a logarithmic scale. Biovolume data were estimated from ECD data provided by ZooProcess, using spherical shape.

A size-based analyses were conducted using PCA (Sect. 2.8) on copepod abundances per size at the different stations, using the size classes defined for the NBSS (Fig. A2). However, for clarity, the size classes were grouped from 15 size classes to just
five, and each class is defined by its ECD instead of its biovolume. These analyses were conducted on the copepod group only. Other taxonomic groups tend to have much larger size ranges, including large organisms such as Chaetognaths or Cnidaria, which introduce significant noise in the NBSS.

## 2.8 Principal Component Analysis (PCA)

PCA was used to evaluate the similarities between the stations based on the abundance of the different taxonomic groups.
Distances between these stations were measured in the phase space of the PCA. Legendre and Gallagher (2001) showed that the Hellinger transformation, prior to PCA, is often preferable to Euclidean distance for calculating distances between samples. Hellinger distance (Rao (1995)) is obtained from:

$$D(x_1, x_2) = \sqrt{\sum_{j=1}^{p} \left( \sqrt{\frac{y_{1j}}{y_{1+}}} - \sqrt{\frac{y_{2j}}{y_{2+}}} \right)^2}, \tag{4}$$

where $p$ denotes the number of categories and $y_{i+}$ is the sum of the frequencies of the ith object.

With this equation, the most abundant species contribute significantly to the sum of squares. The advantage of this approach is that it is asymmetric, meaning it is insensitive to double zeros, which is not the case for Euclidean distance (Prentice (1980); Legendre and Legendre (2012)).





The Hellinger transformation was performed with the labdsv package (Roberts (2023)). The abundance tables were centered and scaled, and the PCA was computed using FactoMineR (Lê et al. (2008)). Prior, to carrying out PCAs, the normality of the data and correlation of the variables were tested, using the Shapiro-Wilk test and Bartlett test of sphericity, respectively.

### 2.8.1 Fixed PCA axis for comparison across layers

To obtain comparable results across depth layers, the PCAs were always conducted in the same way with fixed axes. First, a PCA is performed using data from the 0–400 m layer. Then the datasets from all three layers were projected onto the 0-400 m layer's pair of PCA axes. This approach ensures that comparisons between communities in the three different layers are valid. This was done for community PCA, trophic PCA and size PCA, using matrices of station x groups of size 56 x 8, 56 x 11 and 56 x 5 respectively.

### 2.8.2 Pseudo-F calculation

To quantify the separation of the predefined groups (A, B, F, M) in PCA space, the pseudo-F (Caliński and Harabasz (1974)) is used. This calculation is based on a comparison between inter-group dispersion, measuring variability between groups, and intra-group dispersion, evaluating variability within each group. The pseudo-F calculation is:

$$\text{Pseudo-F} = \frac{\text{Inter-group dispersion}/(k-1)}{\text{Intra-group dispersion}/(n-k)}, \tag{5}$$

where $k$ is the number of groups and $n$ the total number of individuals.

A high pseudo-F value suggests a clear separation between groups, indicating that inter-group variation is predominant compared to intra-group variation.

### 2.8.3 PCA with theoretical $f$ stations

A fundamental question is: is the zooplankton community at the front a mixture of those from water masses A and B, or does it represent a specific community? A method was therefore developed to address this question by creating theoretical $fT$ stations, which have a community composition that is a simple linear combination of those observed at stations $a$ and $b$, as close as possible (i.e., minimal distance) to the observed $f$ stations. The combination of $a$ and $b$ follows the formula:

$$fT = \alpha \cdot a + (1 - \alpha) \cdot b, \tag{6}$$

where $\alpha$ is the proportional contribution from stations $a$ and $b$, respectively. A total of 101 iterations was performed, with $\alpha$ varying from 0 to 1 in increments of 0.01.



Therefore, four new theoretical stations are created per iteration:

$$f1T\_D = \alpha_1 \cdot a1\_D + (1 - \alpha_1) \cdot b1\_D$$

$$f1T\_N = \alpha_1 \cdot a1\_N + (1 - \alpha_1) \cdot b1\_N$$

$$f2T\_D = \alpha_2 \cdot a2\_D + (1 - \alpha_2) \cdot b2\_D$$

$$f2T\_N = \alpha_2 \cdot a2\_N + (1 - \alpha_2) \cdot b2\_N$$

These *fT* stations were then added to the initial datasets. For each of these enriched datasets, a PCA is conducted. After obtaining the coordinates of the stations in the principal component space, the distances between the *fT* stations and their
respective observed *f* stations are calculated for each iteration. For each iteration, the total distance is computed (sum of the distances between the theoretical stations and the original stations), for each transect. Finally, the *fT* station with the minimum total distance is selected, along with the corresponding $\alpha$ value, for each transect. This process enables the generation of new intermediate observations that best reflect the theoretical composition of the front as a linear combination of *a* and *b*.

# 3   Results

## 3.1   Total abundance and biovolume across water masses and layers

The absolute values of abundances and biovolumes of zooplanktonic organisms across different depth layers and stations (Fig. 2) revealed distinct temporal and spatial patterns. In general, abundances and biovolumes of stations within the same water mass decreased over time (stations are presented in chronological order in Fig. 2), with the exception of abundances for the front (Fig. 2.a) and biovolumes for water mass B (Fig. 2.c). The same analysis conducted with the biovolume of organisms < 1
mm (Fig. 2.b) shows strong similarity with figure 2.a based on abundance. When individuals with an ECD greater than 1 mm are included (Fig. 2.c), large differences compared to the total abundances (Fig. 2.a) were observed for all stations.

Regarding the spatial differences during the two front crossings, abundance and biovolume tend to be lower at the front, than in water masses A and B. Indeed, for the first transect, abundances at the front were 67% lower compared to water mass A and 29% lower compared to water mass B. No such decrease was observed for the second transect (Fig. 2.a). Total biovolumes at
the front also showed a decline: 46% relative to water mass A for the first transect and 39% relative to water mass B for the second transect (Fig. 2.c). Interestingly, the second transect reveals greater homogeneity for organisms with a biovolume < 1 mm among water masses compared to the first transect (Fig. 2.b), reflecting the potential influence of post-storm dynamics.

In figure 2.c, strong day-night differences were observed in the vertical distribution of organisms, probably influenced by diel vertical migration (DVM). During the night, between 49 and 62% of the total biovolume was concentrated in the upper
0–100 m layer, except for *b3_N* (due to cnidarians, see next Results section). During the day, between 51 and 83% of the biovolume was in the 200-400m layer, except for *b1_D* with 43%. On average, deep layer biovolume increases by 25% from night to day.







**Figure 2.** a)Total abundance of organisms by intermediate layers and across all sampled stations. b) Total biovolume of organisms, with ESD <1 mm only, by intermediate layers and across all sampled stations. c) Total biovolume of organisms by intermediate layers and across all sampled stations. Stations are in chronological order. The asterisk indicates that the net could not be analysed. Colours refers to the period of the day (blue for midday and black for midnight).





## 3.2 Taxonomic composition across nets and depth layers

The 200 $\mu$m net more efficiently captures copepods, which constitute 45–95% of the relative abundances of taxa, while copepod
abundances comprise only 5-55% in the 500 $\mu$m net (Fig. 3). The larger mesh size is particularly effective for sampling larger
taxa such as Eumalacostraca, Foraminifera, and Cnidaria. The combined samples, which include contributions from both mesh
sizes, still heavily reflect the taxa distributions observed in the 200 $\mu$m net, the abundances of larger organisms sampled with
the 500 $\mu$m net being low. This pattern was also observed in the layers 0-100 m and 0-400 m. Moreover, during the second
transect (after the storm) the dominance of copepods is enhanced in water masses A and F.

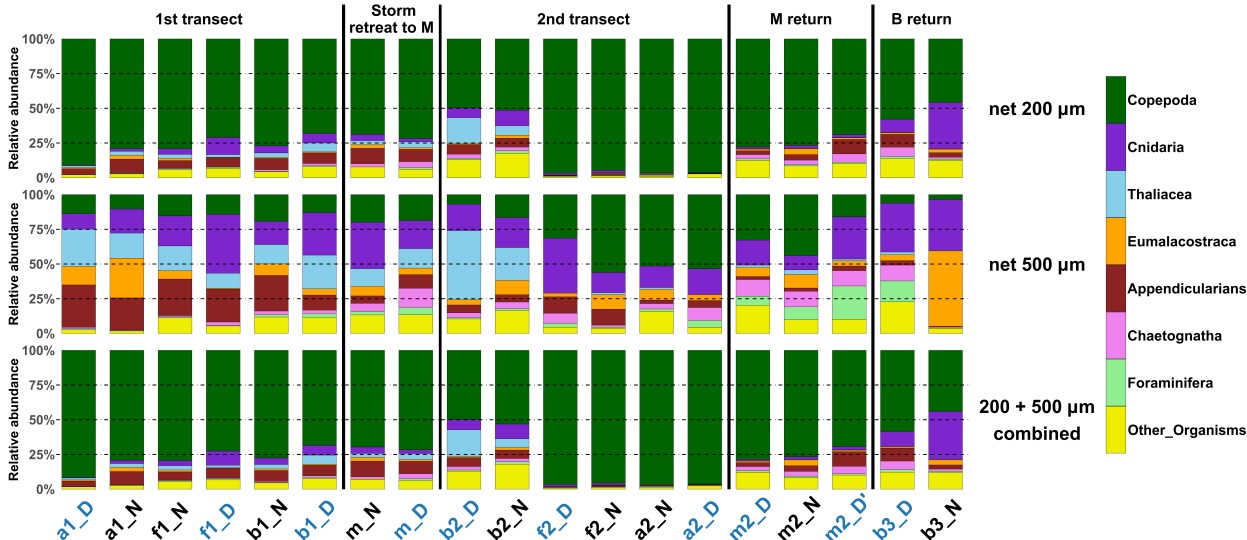

**Figure 3.** Relative abundance of taxonomic groups for nets deployed from the surface to a depth of 200 m, for the two mesh sizes (200
$\mu$m top, and 500 $\mu$m, middle) across all sampled stations (chronological order). Bottom: Relative abundance combining the two mesh sizes.
Colours refers to the period of the day (blue for midday and black for midnight).

In the 0–100 m layer, copepods consistently dominate, comprising at least 45% of the total abundance at nearly all stations
except for *b2* and *b3* (Fig. 4). In the 100–200 m layer there is marked heterogeneity with many stations (8 out of 18) showing
less than 60% copepods. This intermediate layer likely reflects a transitional zone where DVM results in taxonomic shifts. The
200–400 m layer returns to a dominance of copepods at most stations (15 out of 18), with the notable exceptions of station
*b2_N*, where Eumalacostraca account for an anomalously high 55% of the sampled taxa, and *b3_N* where Cnidaria account for
an anomalously high 67% of the sampled taxa.





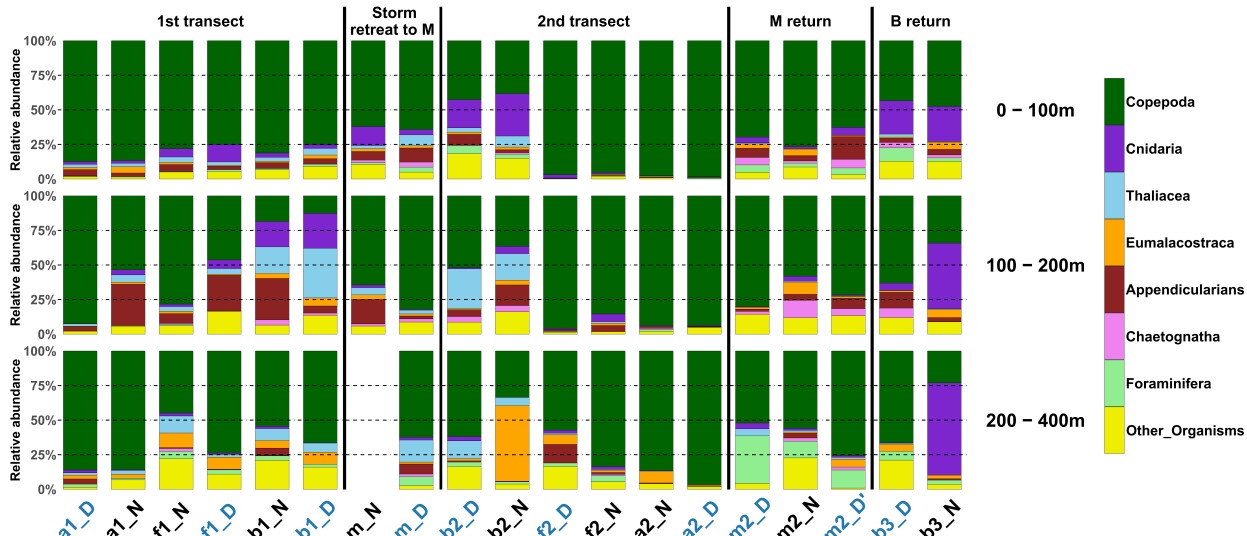

**Figure 4.** Relative abundance of taxonomic groups for the combined nets for the three intermediate layers across all sampled stations (chronological order). The net from station *m_N* at a depth of 400 m could not be analysed. Colour refers to the period of the day (blue for midday and black for midnight).

## 3.3 Diel variations in vertical structuring of zooplankton stocks

Biovolume values show similarities between the 0–100 m and 200–400 m layers which may correspond to a transfer of organisms between day and night layers (Fig. 2.c), whereas the 100–200 m layer (Fig. 4) presents a less structured composition as a potential transition zone for migratory organisms. The analysis of Hellinger distances for the eight taxonomic groups across the three depth layers highlights that the communities in the 0–100 m and 200–400 m layers are more similar to each other, while the intermediate layer (100–200 m) harbours more distinct communities. This pattern was observed across all stations, as well as in separate analyses of the first and second transects. The lowest Hellinger distances were consistently found between the 0–100 m and 200–400 m layers for Copepoda (0.04 and 0.09, respectively), Eumalacostraca (0.03 and 0.08), and Other_organisms (0.06 and 0.03). In contrast, distances involving the 100–200 m layer were approximately four times higher.

Furthermore, the vertical structuring of communities for the two migrant groups, Copepoda and Eumalacostraca, reveals a clear diel pattern: at night, the upper layers (0–100 m and 100–200 m) are more similar, while during the day, the similarity is stronger between the deeper layers (100–200 m and 200–400 m). This pattern reflects a pronounced DVM for both groups with a significant day-night difference (post-hoc, $p < 0.001$ and $0.008$ respectively). Indeed, distances between the 0–100 m and 200–400 m layers were notably higher both during the day (0.24 and 0.13 for Copepoda; 0.48 and 0.38 for Eumalacostraca) and at night (0.29 and 0.34 for Copepoda; 0.38 and 0.52 for Eumalacostraca, respectively for the two transects). In contrast, Hellinger distances between the 0–100 m and 100–200 m layers at night were much lower (approximately 8 times lower for Copepoda and 3 times lower for Eumalacostraca), while during the day the lowest distances were observed between the 100–200 m and 200–400 m layers (approximately 21 times lower for Copepoda and 5 times lower for Eumalacostraca).





## 3.4 Community structure and water mass differentiation

### 3.4.1 Community composition across depths and water masses

PCA_Community summarizes the taxonomic composition of zooplankton communities across water masses and depths (Fig. 5). Axis 1 is inversely correlated to copepod abundance, which stems from the extreme dominance of this group. Axis 2 appears to be more to characteristics of other groups ranging from pure filter feeders (Appendicularians and Thaliacea) to omnivores (Eumalacostraca, Other_organisms) and to carnivores (Chaetognatha and Cnidaria), Formanifera being at the extreme.

In the 0–100 m layer, water masses host distinct communities (Pseudo-F = 3.52). Copepods are more abundant in water masses A and the front, whereas other groups, particularly Foraminifera, Cnidaria, Chaetognatha and Eumalacostraca, dominate in water masses B and M. In this layer, for the first transect, stations *f* are closer to *b* than to *a*. For the second transect, there is a strong similarity between stations *a* and *f*. For the 100–200 m and 200–400 m layers, the taxonomic composition becomes progressively more homogeneous, with reduced differentiation among water masses (Pseudo-F = 1.66 and 1.06).

### 3.4.2 Comparison of the front community composition with adjacent waters

The relative abundances of taxonomic groups across all stations in chronological order are used to compare the community compositions (Fig. 6). Results clearly reveal that the front appears very similar to water mass A in terms of the relative abundance of copepods which progressively decreases from A to F to B. To further investigate these observations, a PERMANOVA was conducted on the abundance of the entire community in water masses A, B, and the front. No significant difference was found between A and F (p = 0.312). However, significant differences were observed between B and F (p = 0.038) and between A and B (p = 0.006). For copepods, significant differences were found between all pairs of water masses and for both transects, as determined by an ANOVA, except between F and A (p = 0.406 for the first transect and p = 0.459 for the second transect). For other groups, significant differences were only observed for Other_organisms between B and A for both transects and between F and B for the second transect.

Figure 7 illustrates the theoretical community distribution at the front, derived from a combination of communities from water masses A and B (Sect. 2.8.3). The positioning of theoretical front stations (*fT*) is displayed within the PCA_Community of figure 5 (Fig. 7.a). For the first transect (Fig. 6.b), the $\alpha$ value (in Eq. 6) is low for the 0-100 m and 200-400 m layer (respectively 0.27 and 0.12) but high for the intermediate layer (0.73). This indicates that the front can be blurred by processes other than just the dynamics of water masses, for instance DVM through the 100-200 m layer. For the second transect, alpha is close to 1, even equal to 1 for the deeper layers, therefore the front is very similar to water mass A (Fig. 7.c).

A notable feature is the position of *fT* stations compared to observed *f* stations within the reduced PCA space. Focusing on the first transect (Fig. 7.b), observed *f* stations appear displaced relative to the *fT* stations. Indeed, in the 0–100 m, 100–200 m layers as well as during the night in the 200-400 m layer, observed *f* stations are positively shifted along axes 1 and/or 2. To examine these shifts, we can reconstitute the theoretical absolute abundance of these *fT* stations and then compare them to the *f* stations. In the 0-100 m the different groups of *fT* are subjected to an average decrease of 35% (with a maximal decrease of 74% in Eumalacostraca). The observed shift is driven by a 72% increase in Cnidaria at the front compared to that is expected





(*fT*), while all other groups exhibit a decline. In the 100-200 m layer the important shift between *f* and *fT* is explained by the substantial increase of 90% in Foraminifera, this group is also the only one showing an increase in this layer (average decrease of 52% for other groups). During the night in the 200–400 m layer, this shift is explained by a pronounced increase of 605% in Chaetognatha (average increase of 84% for other groups). In contrast, the second transect has much higher alpha values, which means a strong similarity between water mass A and F, with a strong domination of copepods in both water masses (Fig. 4). Thus, deviations between *fT* and *f* are very low and could not be analyzed.

### 3.4.3   Trophic composition and dynamics

PCA_Trophic (Fig. 8) was conducted based on the abundance of each trophic category of Table 2. In the 0–100 m layer, non-carnivorous groups (detritivores and herbivores) dominate in water masses A and F, while carnivorous groups are more abundant in water masses B and M. Indeed, the PCA_Trophic reveals a clear trend in the distribution of trophic groups, with carnivores predominantly located in a quarter plan extending from the axes of Cop_CCF to Cni, while herbivorous and non-carnivorous groups are situated on the rest of the plan (from Cop_FF to App). Whereas the community in *a* stations is clearly dominated by non-carnivorous in 0-100 m and 200-400 m, a shift towards carnivory is observed in the 100–200 m layer for these stations. The *b* and *m* stations are more dominated by filter-feeders and predators (Cni and Cha) in surface.

In the 200-400 m layer all the stations becoming less distinctly grouped and converging closer to the PCA center. In this layer, a positive shift along the x-axis is also observed. Upon closer examination, this shift is not driven by a high contribution of non-carnivorous copepods, but rather by a high contribution of Eumalacostraca and Foraminifera, and a low contribution of Chaetognatha. The discrimination of the stations among the water masses is stronger (approximately three times greater) for the 0-100 m layer compared to the other layers, indicating that, in terms of trophic functions, the water masses are less differentiated for these deeper layers. Stations within each water mass appear relatively similar to one another, with the exception of station *a1_N* at 100-200 m, which stands out due to a high abundance of copepods carnivores ambush-feeders and Appendicularia.

### 3.4.4   Size composition of copepods

In the 0–100 m layer, water masses exhibit the strongest differentiation in terms of copepod size structure, with *a* and *f* stations characterized by larger copepods (>1500 $\mu$m; Fig. 9). The *b* and *m* stations, on the other hand, are characterized by smaller copepods with *b* predominantly ranging from 300 to 450 $\mu$m and *m* from 450 to 950 $\mu$m. Meanwhile, the PCA_Size shows a much more heterogeneous distribution for the *b* stations. As depth increases, size composition becomes more homogeneous, with all stations clustering near the PCA centre, but slightly shifted toward highest size. Indeed, there is a decrease in Pseudo-F with depth, respectively 4.47, 1.46 and 0.71. This concentration near the PCA centre and the decrease in Pseudo-F indicate a gradual decrease in variability among the deep stations, i.e., the differences between stations become less pronounced. This is also observed in the three previous PCAs (Fig. 5, 7, 8), but it is more pronounced here for the copepod size composition.



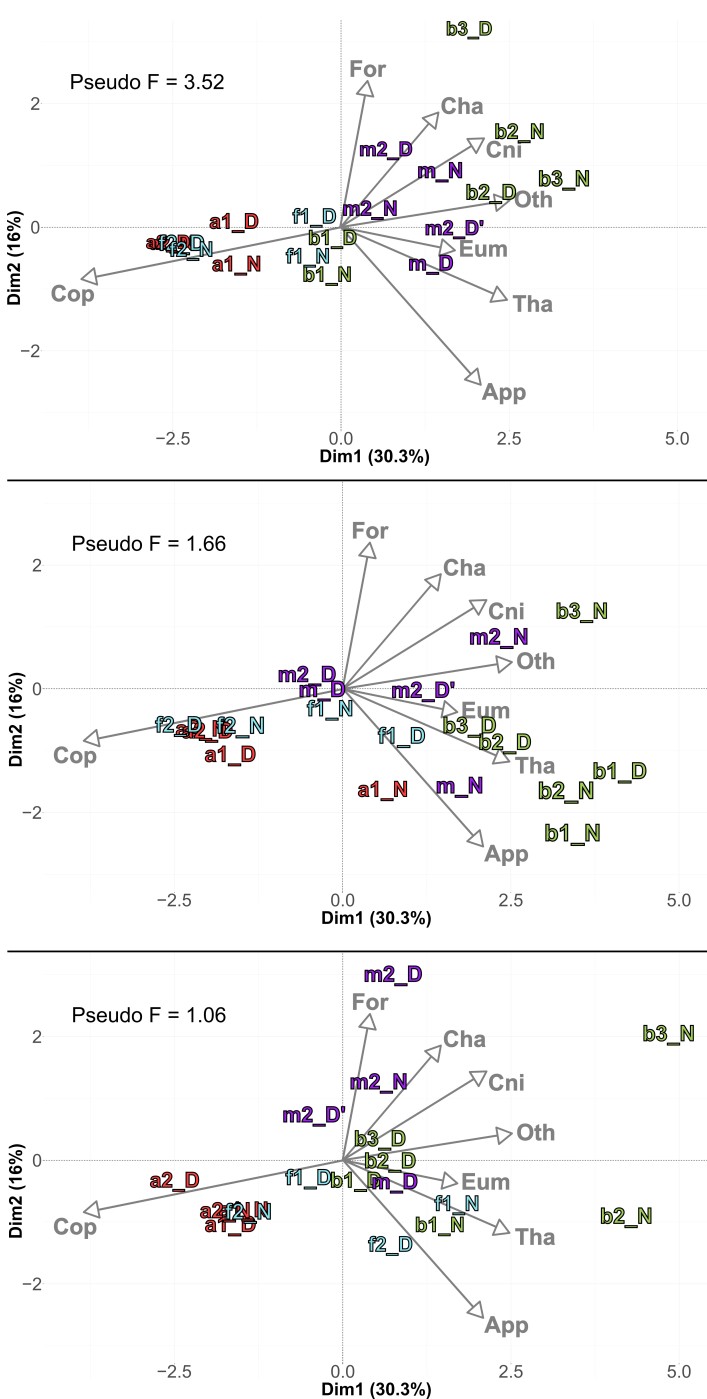

**Figure 5.** PCA_Community illustrating the composition of communities, based on relative abundance data (Hellinger transformation) from all stations for each intermediate layers. The axis computed for 0 – 400 m were used for the three layers. Colour refers to the water mass (red for A, green for B, cyan for F and violet for M). In 0-100 m: stations *a2_D*, *a2_N*, *f2_D* and *f2_N* overlap at dim1 = -2.3 and dim2 = -0.3. In 100-200 m: stations *a2_D*, *a2_N*, *f2_D* and *f2_N* overlap at dim1 = -2 and dim2 = -0.7. In 200-400 m: stations *a1_D*, *a1_N*, *a2_N* and *f2_N* overlap at dim1 = -1.7 and dim2 = -1.





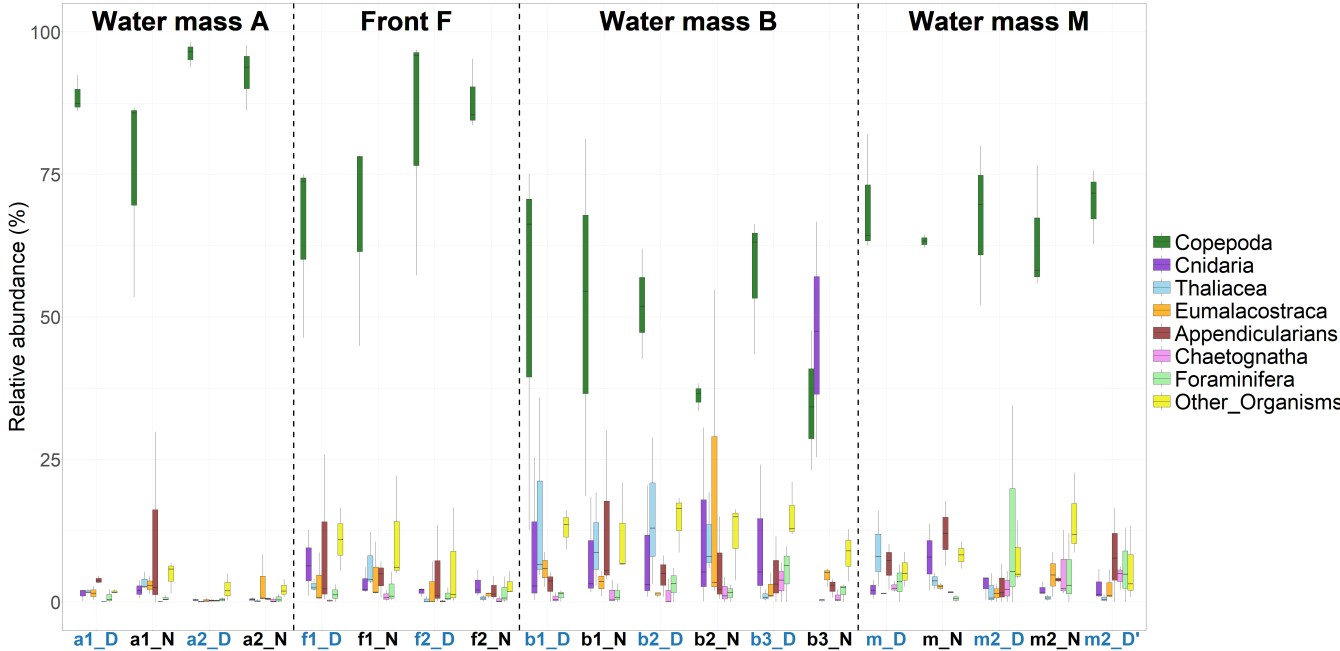

**Figure 6.** Relative abundance of taxonomic groups across all stations, presented as a boxplot. Sorted by water mass and incorporating data from all three layers of each station. Colour refers to the period of the day (blue for midday and black for midnight).





**Figure 7.** a) PCA_Community illustrating the composition of communities, based on relative abundance data (Hellinger transformation) from all stations for each of the intermediate layer (same as figure 5). The closest theorical $fT$ of each observed $f$ is plotted, with the corresponding $\alpha_1$ and $\alpha_2$ values of each $fT$'s couple for the 1st and 2nd transect, respectively. b) Zoom for the stations of the 1st transect. c) Zoom for the stations of the 2nd transect. In c), in 0-100 m stations $a2\_D$, $a2\_N$, $f2\_D$, $f2\_N$, $f2T\_D$ and $f2T\_N$ overlap at dim1 = -2.5 and dim2 = -0.2. In 100-200 m stations $a2\_D$, $a2\_N$, $f2\_D$, $f2\_N$, $f2T\_D$ and $f2T\_N$ overlap at dim1 = -1.8 and dim2 = -0.6. In 200-400 m stations $f2T\_D$ and $a2\_D$ overlap at dim1 = -2 and dim2 = -0.2; $a2\_N$, $f2\_N$ and $f2T\_N$ overlap at dim1 = -1 and dim2 = -0.4.




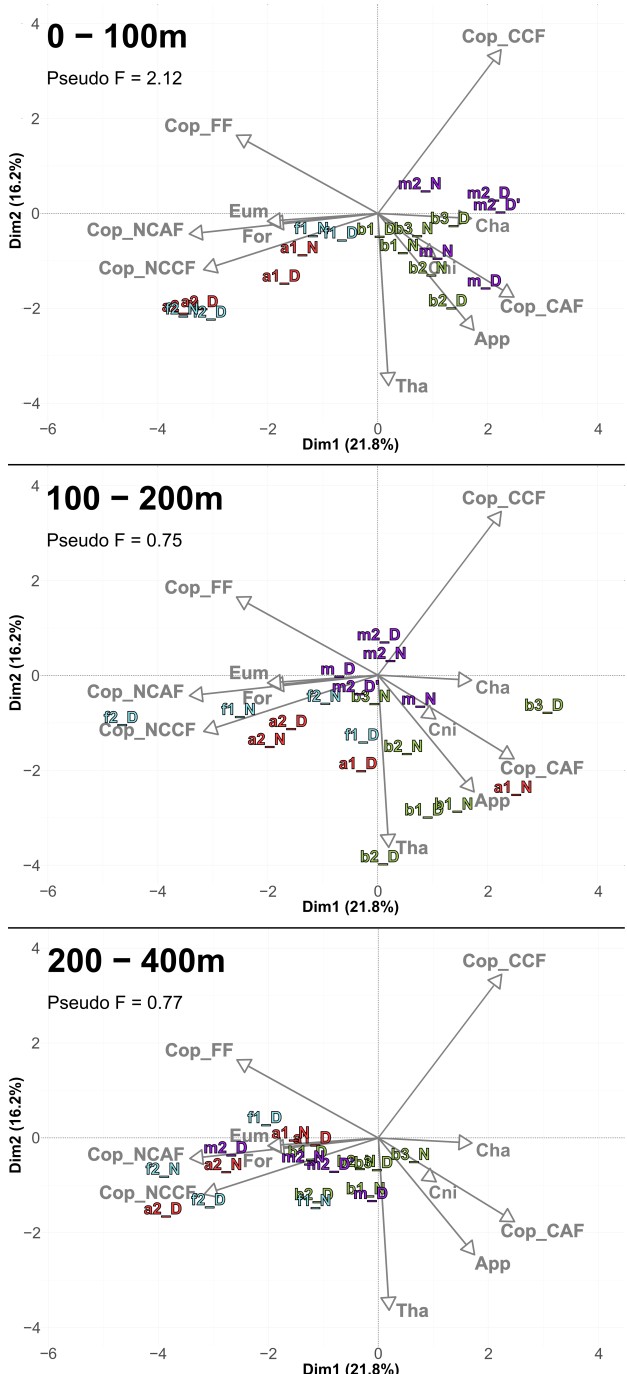

**Figure 8.** PCA_Trophic illustrating the trophic composition of communities, based on relative abundance data (Hellinger transformation) from all stations for each intermediate layer. The axis computed for 0 – 400 m were used for the three layers. Colour refers to the water mass (red for A, green for B, cyan for F and violet for M). In 0-100 m: stations *a2_D*, *a2_N*, *f2_D* and *f2_N* overlap at dim1 = -3.5 and dim2 = -2. In 200-400 m: stations *a1_D* and *a1_N* overlap at dim1 = -1.5 and dim2 = 0; stations *b1_D*, *m2_N*, *m2_D*, *b2_N* and *b3_D* overlap at dim1 = -1 and dim2 = -0.5; *b2_D* and *f1_N* overlap at dim1 = -1.3 and dim2 = -1.3; *b1_N* and *m_D* overlap at dim1 = -0.1 and dim2 = -1.3.





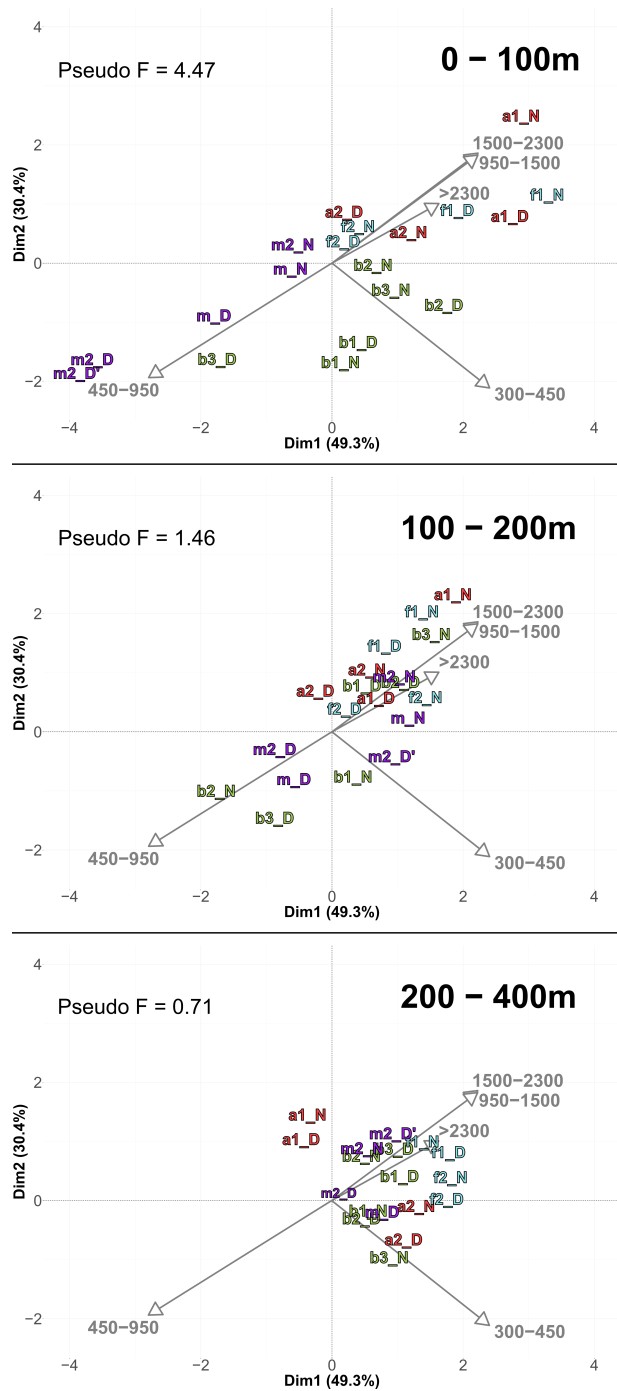

**Figure 9.** PCA_Size illustrating the body size composition of copepods, based on relative abundance data (Hellinger transformation) from all stations for each intermediate layer. The size classes (in $\mu$m) were defined according to those from NBSS. The axis computed for 0 – 400 m were used for the three layers. Colour refers to the water mass (red for A, green for B, cyan for F and violet for M). In 100-200 m: stations *a2_N*, *b1_D*, *b2_D* and *m2_N* overlap at dim1 = 0.8 and dim2 = 0.9. In 200-400 m: stations *b2_N*, *m2_N*, *b3_D*, *m2_D'* and *f1_N* overlap at dim1 = 1 and dim2 = 0.9; *b1_N*, *b2_D* and *m_D* overlap at dim1 = 0.6 and dim2 = -0.3.





# 4  Discussion

## 4.1  Zooplankton biovolumes, abundances and community structure across water masses

The spatial differences between water mass A and B in late spring can be linked to the regional hydrological and ecosystem
functioning of the NWMS in the post-bloom period (D'Ortenzio and Ribera d'Alcalà (2009)). Water mass A has its origin in
the Liguro-Provencal area (NWMS), characterized by intense convection and mixing (Barral et al. (2021)) and by high nutrient
concentrations associated with strong gradients and shallower nutricline depths (Severin et al. (2017); Joël et al., 2025 in prep.).
In the transitional post-bloom period (April-May), water mass A is still nutrient-rich (Severin et al. (2017); Joël et al., 2025 in
prep.) and productive (Mayot et al. (2017); Hunt et al. (2017)) with the formation of a deep chlorophyll maximum around 50
m (Fig. A3; Lavigne et al. (2015); Doglioli et al. (2024)). Water mass A shows higher zooplankton stocks strongly dominated
by copepods and larger forms (Fig. 4, 9). Water mass B in the southern part of the NBF comes from the epipelagic waters
of the Algerian basin, which are warmer and fresher than waters from the NWMS, with virtually permanent stratification
and a DCM deeper than 50 m (Fig. A3; Lavigne et al. (2015)). It is associated with lower nutrient concentration gradients
and deeper nutricline depths (Joël et al., 2025 in prep.). In water mass B, zooplankton stocks fluctuate little over the year
(Fernández de Puelles et al. (2004)), and community structure is dominated by small sizes and slightly more diversified within
the non-copepod organisms (Fig. 4, 9).

Mesozooplankton data from the two transects across the NBF during the BioSWOT-Med campaign can only be compared
with a very limited number of other observations, particularly in the vicinity of the front. The two DEWEX campaigns (Conan
et al. (2018)), in February and April 2013, studied dense water formation and the pelagic ecosystem's biogeochemical and
ecological responses, including zooplankton mapping during the winter-spring transition (Donoso et al. (2017)). The stations
of water mass A were in the Deep Convection Zone (DCZ) near the LION Station (42°04′ N, 4°38′ E; Margirier et al. (2020)),
whereas the nearest station of water mass B was situated north of Menorca Island (Donoso et al. (2017)). In winter, the DCZ
exhibited its lowest zooplankton abundance and biomass in the top 200 m (200 ind/m$^3$ and 5 mg DW/m$^3$), in contrast to the
periphery of the DCZ and especially the Balearic stations (650 ind/m$^3$ and 10 mg DW/m$^3$). In spring, this pattern reverses
with higher zooplankton stocks in the DCZ (4400 ind/m$^3$ and 100 mg DW/m$^3$) and lower in the periphery, including the
Balearic stations (2000 ind/m$^3$ and 30 mg DW/m$^3$). During BioSWOT-Med, water mass A exhibited approximately twice
the abundances and biovolumes compared to water mass B and the frontal zone F during the first transect (Fig. 2). However,
during the second transect, abundances and biovolumes in water mass A were much lower compared to the first transect. This
overall decrease during BioSWOT-Med might be explained by the 10 day interval between the two transects in a phase of
seasonal decline in zooplankton stocks in the convection zone (Berline et al. (2011); Auger et al. (2014)). Additionally, the
storm occurring between the two transects could have influenced these decreases.

Our biomasses in water mass A match the biomasses observed in the central part of the NWMS (converting our biovolumes
to biomass using DW/WW of 10%, and 1 mg WW equal to 1 mm$^3$). Biomasses in water mass B align with values observed
around the Balearic Islands and in the Algerian Basin. A similar comparison also applies to abundances (Nival et al. (1975);
Siokou-Frangou et al. (2010); Nowaczyk et al. (2011); Donoso et al. (2017); Fierro-González et al. (2023); Fernández de





Puelles et al. (2023)). Another interesting pattern is the highest percentage of copepods in mesozooplankton found in water

mass A (around 85%) compared to water mass B (around 40%), which is consistent with other observations made in the same
water masses (Nowaczyk et al. (2011); Feliú et al. (2020); Fierro-González et al. (2023); Fernández de Puelles et al. (2023);
Siokou et al. (2019)). However, these previous observations in the NWMS were not dedicated to the NBF.

In the NWMS, studies dedicated to the impact of frontal dynamics on the stocks and structure of mesozooplankton have
been carried out in the front associated with the northern current in the Ligurian (Prieur and Sournia (1994); Boucher et al.
(1987); Panaïotis et al. (2024)) and the Catalan seas (Alcaraz et al. (2007); Saiz et al. (2007)). In both areas, the front does
not appear to be an area of higher zooplankton biomass than adjacent waters (Boucher et al. (1987); Panaïotis et al. (2024);
Alcaraz et al. (2007); Saiz et al. (2007)), but it is the site of higher physiological rates (spawning rates, larval growth) of the
organisms (Boucher et al. (1987); Alcaraz et al. (2007); Saiz et al. (2007)), enhanced by higher prey densities and turbulence
levels (Alcaraz et al. (2007)). Studies of species distributions have shown that the front associated with the northern current
can also represent a barrier for coastal species in their distribution to the central zone (Pedrotti and Fenaux (1992); Saiz et al.
(2014)), or conversely, from the central zone to the coast (Berline et al. (2013)).

## 4.2    Complexity of concurrent processes impacting zooplankton biomass distribution at front

The decline in zooplankton abundances and biovolumes at the front during the first transect, and the weak increase in abundance
but not in biovolume during the second transect (Fig. 2) could reflect specific hydrological and physical mixing characteris-
tics of the northern Balearic front (Salat (1995); Alcaraz et al. (2007)), where dynamic turbulence and horizontal dynamics
appeared less favourable for biomass accumulation. Actually, while turbulence at fronts is known to enhance nutrient diffu-
sion to phytoplankton, promoting enriched food webs for zooplankton (Kiørboe (1993); Estrada and Berdalet (1997)), it can
also increase encounter rates between particles and consumers, influencing community interactions (Rothschild and Osborn
(1988); Alcaraz et al. (1989); Saiz et al. (1992); Caparroy et al. (1998)). Indeed, the front in our area of interest, sampled
by Lagrangian drifters at 1 and 15 m depth (Demol et al. (2023)), showed prevailing along-front deformation and patches of
water mass convergence and divergence inducing variable vertical velocities up to approximately +/- 1 mm/s in the upper 15
m sea layer (Berta (2025)). Moreover, the core of the front, as identified by ADCP transects (Petrenko et al. (2024)), is found
within 100 m depth and 20 km width. Consequently, considering the frontal spatial scales, as well as the divergence and the
vertical transport magnitude and variability, we expect that our results do not reveal significant effects beyond 100 m depth
and that mixing has shorter time scales compared to zooplankton development times (several weeks to months). In a study of
154 glider-resolved fronts across the California Current System, Powell and Ohman (2015a) found that zooplankton biomass
was often, but not always, enhanced, also indicating variations in matchup of frontal duration and zooplankton development
time. Finally, our campaign took place in late April to early May, corresponding to the post-bloom period, when phytoplankton
biomass levels are already too low to sustain optimal growth of specific zooplankton groups.

        The observed variability in zooplankton abundances and biovolumes over time and space (Fig. 2) underscores the complexity
of concurrent processes acting at different scales, such as DVM or storm events that interact with the hydrological processes



creating the front. Furthermore, biomass peaks heavily depend on the taxa considered (Mangolte et al. (2023)) and our analyses
never consider a single taxon but always groups of organisms (Table 2) or even the whole sampled mesozooplankton.

## 4.3   Investigating the front: mixing zone or distinct community?

A fundamental question in this study was whether the front is a mixture of communities from water masses A and B, or if
it hosts a distinct community with notably different abundances, or even the presence of new taxa. Our results reveal clearly
that the front appears very similar to water mass A in terms of the relative abundance of copepods, with decreasing relative
abundance from A to F to B (Fig. 6). Moreover, in the 0-100 m layer, the shifts between the projections of $f$ and $fT$ (Fig. 7)
suggests a less disruptive influence of the front on Cnidaria and Foraminifera. In contrast, the pronounced decrease in Thaliacea
may indicate active avoidance of the frontal region.

We note that the primary differences among higher taxa across the front concern not the most abundant groups, but secondary
groups: Cnidaria, Foraminifera and Eumalacostraca for 0-100 m; Cnidaria and Foraminifera for 100-200m. This indicates that
the composition of zooplankton communities is shaped by species-specific responses, highlighting the importance of consid-
ering individual taxonomic groups rather than just overall abundance patterns when analysing community dynamics. In other
frontal studies, some taxa have been found more abundant than in adjacent waters, showing that zooplankton abundance at
fronts is species-specific (Molinero et al. (2008)). Gastauer and Ohman (2024) found that appendicularians, copepods, and
rhizarians sometimes showed front-related increases in concentration. These patterns likely result from interactions between
species-specific behaviors and frontal dynamics. Indeed, certain zooplankton can maintain their depth through vertical swim-
ming or buoyancy control, allowing them to counteract vertical currents and form aggregations at the front (Franks (1992)). As
a result, fronts can enhance abundance or facilitate exchanges for particular species depending on their behavioral strategies
(Wishner et al. (2006)).

To answer our initial question, the results suggest that for the first transect, the front is indeed a mix of A and B communities,
but it also shows higher abundances of organisms such as Cnidaria, Foraminifera and Chaetognatha, which are less represented
in A and B. For the second transect, the storm of the previous days may have altered the community structure (a hypothesis
that will be further discussed in Sect. 4.5), making it difficult to draw definitive conclusions. It can also be noted that no
taxon was found exclusively at the front in either transect. We note that our more detailed analysis revealed the absence of a
few low-abundance taxa at the front compared to water masses A or B (e.g., Magelonidae, cyphonautes, echinoderm larvae,
radiolarians, and Heteronemertea).

## 4.4   Community structure with depth

The zooplankton community reveals clear vertical stratification, with distinct compositions at different depths. Zooplankton
taxonomic composition becomes more homogeneous with depth between water masses A, B, and the front, as the influence
of the front weakens below 100 m and becomes negligible below 200 m. The deep zone below the front reflected reduced
differentiation among water masses and weakening hydrological influence (Fig. 5, 8, 9), as generally observed (Scotto di Carlo





et al. (1984)). Thus, the higher discrimination between water masses in the upper 0-100 m layer emphasizes the stronger influ-ence of hydrology and biological productivity at the surface. Besides physical processes (Sect. 4.2), differences in zooplankton taxonomic and size distributions between water masses A, B, and the front are influenced by phytoplankton size structure and trophic interactions with nano- and micro-grazers (Bănaru et al. (2014); Hunt et al. (2017); Tesán Onrubia et al. (2023)).
Ongoing analyses will explore the food-web pathways to zooplankton in these water masses (Bănaru et al., in prep.; Joël et al., 2025 in prep.). Our findings (Fig. 8, 9) already highlight the dominance of non-carnivorous, large copepods in water masses A and F.

The zooplankton structure in the surface layer was also clearly affected by the behaviour of migrating zooplankton. Our results clearly showed that the taxonomic and size distributions of migrant zooplankton were closer between layers of 0-100
m and 100-200 m at night and between layers of 100-200 m and 200-400 m during the day (Sect. 3.3, Fig. 4) which is a direct consequence of the population movements that make up DVM. The copepod and eumalacostracan groups present in our samples are well documented as performing DVM (Andersen and Sardou (1992); Isla et al. (2015); Guerra et al. (2019)). This deep DVM observed in our samples is likely driven by species of *Euchaeta*, *Pleuromamma*, and *Heterorhabdus*, as Andersen et al. (2001b) demonstrated their DVM amplitude can reach up to 400-500 m for some taxa.

## 4.5   Storm impact?

The Provençal basin in NWMS is an area highly exposed to strong regional winds (i.e., "mistral" and "tramontane") blowing southwards, with a marine origin in the Gulf of Lion that is particularly frequent in autumn and winter. Such winds can, nonetheless, occur more episodically but intensely in spring and even summer (Morales-Márquez et al. (2021)). During an intense two-day wind-storm in the Ligurian Sea, Barrillon et al. (2023) observed a significant hydrological effect on the upper
60 m, impacting nutrient supply and chlorophyll biomass over the following 24 hours. However, the influence on zooplankton was not documented. Andersen et al. (2001a) sampled two one-day wind-storm events in the Ligurian Sea. The main effects also occurred in the upper surface layer. The main observed impact on the zooplankton was an increased production of nauplii linked to adult spawning, but a decrease of zooplankton biomass (mainly copepods). Analyzing the zooplankton vertical distributions from Niskin bottles sampling , Andersen et al. (2001b) noted upward aggregation within the upper 40 m of naupliar stages of
copepods and euphausiids as well as small-sized copepods. They do not mention any effect on other groups than crustaceans, notably gelatinous plankton, from either bottle samples or from plankton nets.

During BioSWOT-Med, a storm (NW winds) peaked on May 2nd, between the first and the second transect. The two transects were separated by nine days and : ~50 km. The observed changes in community structure and distribution between the two transects may have resulted from the mixing and dilution associated with the storm, but also from patchiness or temporal
evolution. However, we observed a limited influence on mixed layer depth (deepening from ~15 m to 30 m) and also on chlorophyll-a fluorescence profiles, according to glider data, similar to the observations by Barrillon et al. (2023) and Andersen et al. (2001b). Comparison of abundances between the first to the second transect revealed significant differences for the 0–100 m layer, but not in deeper layer, therefore potentially linked to the storm (Table 3). In the surface layer, small and mid-sized





copepods were the most affected, but also chaetognaths and cnidarians. Large migrant copepods, such as *Pleuromamma* and

*Euchaeta*, appeared weakly affected. A similar trend was observed for Calanoida, which includes both small and large, migrant and non-migrant species. Analyses of the whole planktonic community response to the storm (including phytoplankton) are required to better understand the observed zooplankton changes.

**Table 3.** Results of ANOVA tests (H0: no differences of averages between the first and the second transect) performed on the eight taxonomic groups and copepod subgroups (eight most abundant species). For each significant ANOVA result (p < 0.05), a Tukey's Honest Significant Difference test was applied to identify differences between the first and the second transect for each water mass (shown in the last four columns). For layers 100-200 m and 200-400 m, no significant differences were found.

| Type of analysis | Depth | Taxonomic group / species | ANOVA p-value | p-value A1st vs A2nd | p-value B1st vs B2nd | p-value F1st vs F2nd | p-value M1st vs M2nd |
|---|---|---|---|---|---|---|---|
| ANOVA Depth | 0-100 m | Appendicularians | 0.124 | | | | |
| | | Chaetognatha | **0.039 \*** | **<0.001 \*\*\*** | 0.0659 | **<0.001 \*\*\*** | 0.108 |
| | | Cnidaria | **<0.001 \*\*\*** | **<0.001 \*\*\*** | **<0.001 \*\*\*** | 0.418 | 0.765 |
| | | Copepoda | **<0.001 \*\*\*** | 0.189 | **<0.001 \*\*\*** | **<0.001 \*\*\*** | 0.617 |
| | | Eumalacostraca | 0.534 | | | | |
| | | Foraminifera | 0.429 | | | | |
| | | Other_Organisms | 0.375 | | | | |
| | | Thaliacea | 0.929 | | | | |
| ANOVA Copepod species | 0-100 m | Calanoida | 0.255 | | | | |
| | | Centropages Typicus | **0.014 \*** | **0.002 \*\*** | 0.104 | **< 0.001 \*\*\*** | 1 |
| | | Maxillopoda | 0.052 | | | | |
| | | Corycaeidae | **0.0104 \*** | **< 0.001 \*\*\*** | 0.797 | **< 0.001 \*\*\*** | 0.992 |
| | | Euchaeta | 0.581 | | | | |
| | | Oithona | **0.0231 \*** | 0.448 | **0.0197 \*** | 0.923 | 0.876 |
| | | Oncaeidae | **0.015 \*** | **< 0.001 \*\*\*** | **0.031\*** | **0.025 \*** | 0.87 |
| | | Pleuromamma | 0.928 | | | | |

# 5 Conclusion

To our knowledge, this study represents the first detailed investigation of the fine-scale zooplankton distribution of the North

Balearic Front in late spring, linking finescale dynamics to mesozooplankton distributions. Our findings reveal that the North Balearic Front exhibits characteristics more akin to a boundary between water masses than a zone of pronounced biological accumulation.

Key observations include the stratified vertical distribution of zooplankton communities, with distinct taxonomic compositions in the surface, intermediate, and deeper layers, and a progressive homogenization of community structure with depth.

DVM was particularly evident, underscoring the dynamic nature of zooplankton behaviour in relation to environmental gradients. Moreover, post-storm analyses highlighted the susceptibility of these communities to episodic weather events, which can disrupt established ecological patterns.

These results challenge generalized assumptions about the ecological role of oceanic fronts. They underscore the importance of high-resolution observations across horizontal and vertical spatial scales, temporal processes ranging from days to weeks,

and precise taxonomic determination to fully understand the complexity of mesozooplanktonic communities in frontal zones.





Further trophic studies based on stable isotope ratios and the biochemical composition of zooplankton and phytoplankton size classes are still needed to decipher the complexity of trophic interactions in the two adjacent contrasting zones and in the frontal zone under the effect of nutrient input determined by physical processes. In addition, our net sampling approaches need to be complemented by continuous measurement techniques, such as autonomous gliders, bioacoustics and satellite data, with
in-situ sampling to better capture the spatial and temporal variability of these systems. This approach would enable a more comprehensive assessment of how physical and biological processes interact to shape zooplankton communities at oceanic fronts.

## Data Availability Statement

The Sentinel-3 data used in this manuscript are available and freely accessible to the public (https://www.copernicus.eu/en),
accessed on 11 November 2024.

## Author contribution

Maxime Duranson was responsible for data curation, formal analysis, visualization, and conceptualization. François Carlotti and Léo Berline contributed to conceptualization and were responsible for supervision and validation. Loïc Guilloux performed the ZooScan processing and taxonomic identification of the samples. Maxime Duranson prepared the manuscript with review
and editing of all co-authors.

## Acknowledgments

The authors thank the TOSCA program of the CNES (French Spatial Agency) which funds the BIOSWOT-AdAC project and the ANR – FRANCE (French National Research Agency) for its financial support to the BIOSWOT ANR-23-CE01-0027 project.
The FOF (French Oceanographic Fleet) and, in particular, the captain Gilles Ferrand and the crew of the R/V L'Atalante are acknowledged for their precious support during the BioSWOT-Med cruise.

M.D. Ohman and S. Gastauer were supported by U.S. National Science Foundation grant OCE-2243190.

Alice Della Penna was supported by the University of Auckland via the Faculty Research Development Fund (Grant 3724591)

Dr. Maristella Berta contribution was supported by the ITINERIS Project (IR0000032-Italian Integrated Environmental Research Infrastructures System-CUP B53C22002150006) and by CNR-ISMAR (Lerici, Italy) dedicated fundings.



# Appendix

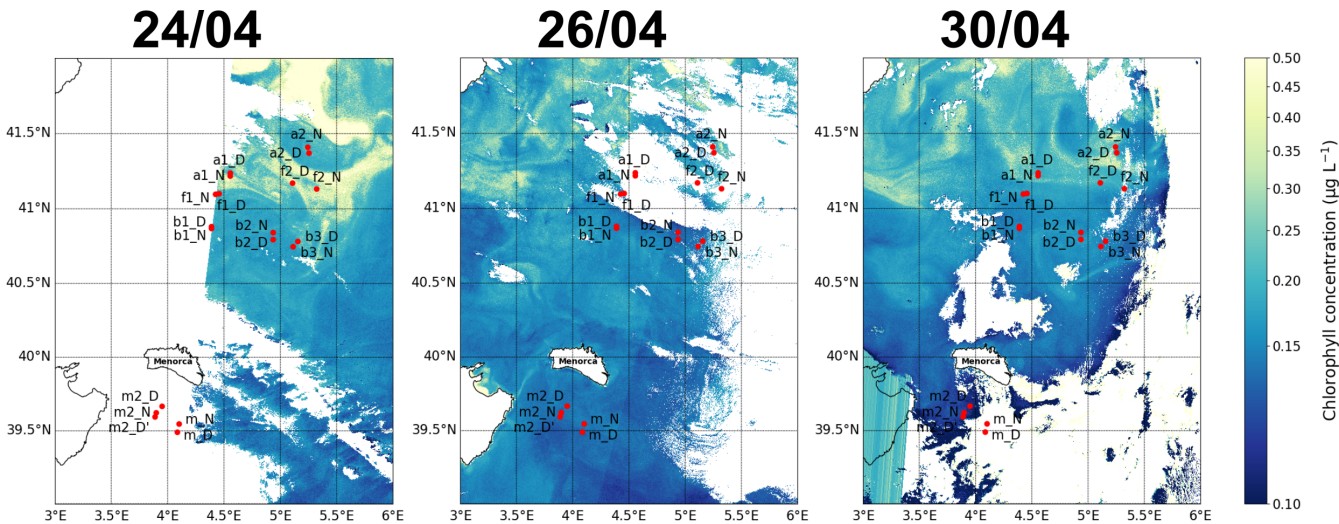

**Figure A1.** Maps of the sampling stations with surface chlorophyll concentration for 3 different days (as complement of Figure 1).

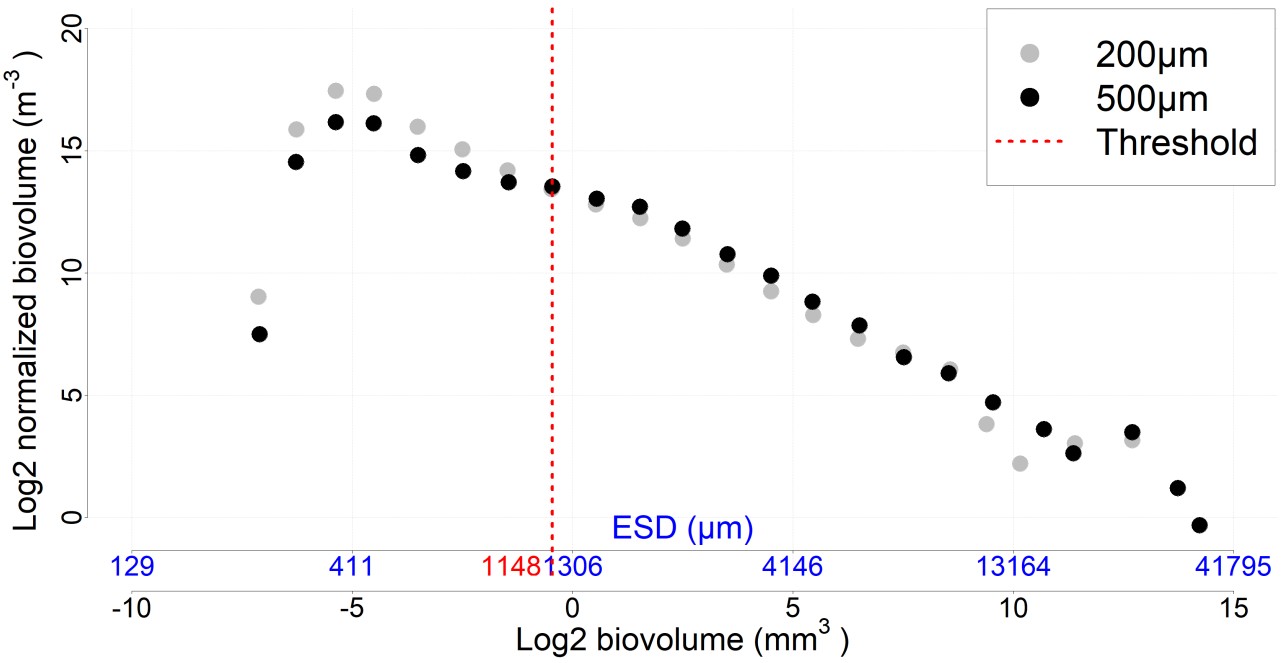

**Figure A2.** NBSS incorporating all data (all stations and depths) for each mesh size. The threshold value represents the organism size above which the 500 $\mu$m nets sample more efficiently than the 200 $\mu$m nets.





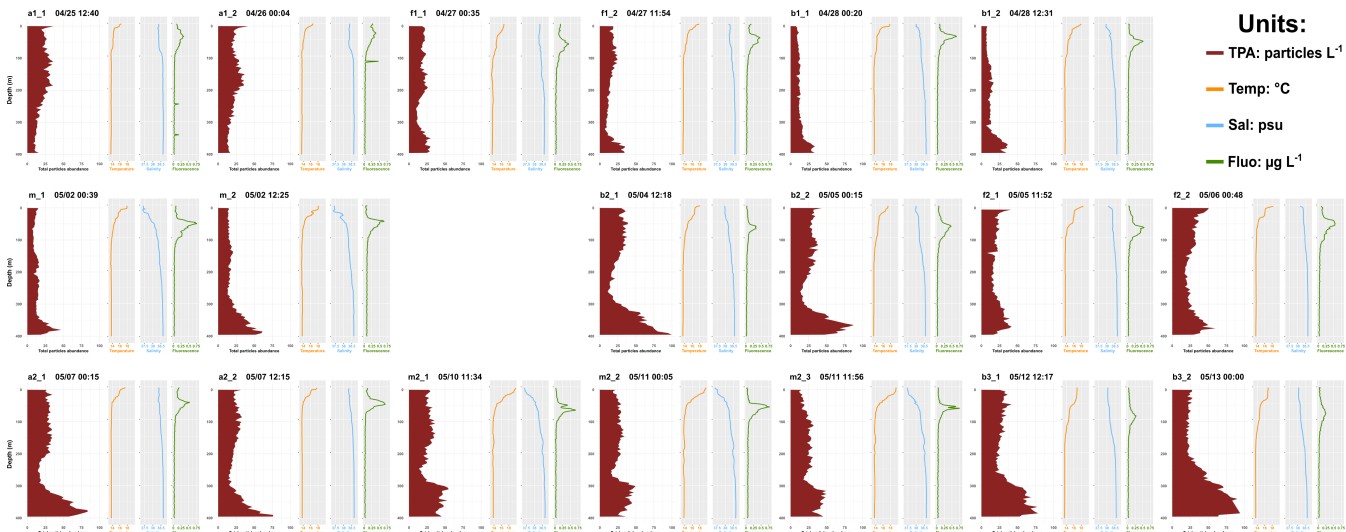

**Figure A3.** Total particles abundance, temperature, salinity, and fluorescence profiles.



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
