# Peer review of "The North Balearic Front as an ecological boundary: zooplankton fine-scale distribution patterns in late spring"

_EGUsphere, 2025_

## Author Comment (AC1)

We would like to thank the reviewer for the thorough and constructive comments. These suggestions have been carefully considered and have helped us to improve the manuscript. Below, we provide a detailed response to each comment.

**General comments:**

-Missing taxa: Clausocalanus and Ctenocalanus, the first being a dominant genus in the open Mediterranean Sea and the latter also co-occurring commonly and with high relative abundance (e.g., Siokou-Frangou et al., 2010). I do not see these genera in any of the categories considered in Table 2. This is an important problem that must be addressed and solved.

The semi-automatic taxon recognition process is done on images from Zooscan, with pixel size of 10.3 µm. Therefore, some taxa can be identified down to species level, but other are only identified at genus, family or order. The issue is that the majority of calanoid copepods have not been identified beyond the order. So Clauso- and Ctenocalanus are not missing in the counts, but they are pooled with other calanoids, in the 'undetermined Calanoida' (around 50% of total abundance) group, which is then not mentioned in Table 2.

A new version of the Table 2 has been now defined. This does not solve the above-mentioned limitation, which is inherent to the use of semi-automatic taxonomic recognition in EcoTaxa. However, the revised table avoids possible confusion by showing only representative taxa, those that account for at least 1 % of the total abundance within that specific category.

| Category         | Abbreviation | Representative Taxonomic Group identified by ZooScan                                                                                                                                                                                     |  |  |  |
|------------------|--------------|------------------------------------------------------------------------------------------------------------------------------------------------------------------------------------------------------------------------------------------|--|--|--|
| Appendicularians | Арр          | Oikopleuridae, Fritillariidae, Appendicularia undetermined                                                                                                                                                                               |  |  |  |
| Chaetognatha     | Cha          | Chaetognatha undetermined                                                                                                                                                                                                                |  |  |  |
| Cnidaria         | Cni          | Cnidaria (ephyra), Hydrozoa, Siphonophorae, Physonectae, Trachylinae (Aglaura, Solmundella), Diphyidae                                                                                                                                   |  |  |  |
| Copepoda         | Сор          | Calanoida undetermined, Oithona, Centropages (Centropages typicus, Centropages undetermined), Oncaeidae, Pleuromamma (Pleuromamma undetermined, Pleuromamma abdominalis), Corycaeidae (Corycaeidae undetermined, Urocorycaeus), Euchaeta |  |  |  |
| Eumalacostraca   | Eum          | Euphausiacea larvae, Amphipoda ( Phronima , Amphipoda undetermined, Hyperiidae undetermined), Eumalacostraca undetermined, Decapoda (Dendrobranchiata), Euphausiacea undetermined                                                 |  |  |  |
| Foraminifera     | For          | Foraminifera undetermined                                                                                                                                                                                                                |  |  |  |
| Thaliacea        | Tha          | Doliolida, Thaliacea undetermined, Salpida (Salpida undetermined, Salpa fusiformis)                                                                                                                                                      |  |  |  |
| Other Organisms  | Oth          | Limacinidae, Ostracoda, Errantia, Pteropoda (Pteropoda undetermined, Cymbuliidae), Crustacea (Crustacea undetermined, nauplii)                                                                                                           |  |  |  |

**Explanatory paragraph included in the manuscript:**

The semi-automatic taxon recognition process was performed on Zooscan images with a pixel size of 10.3 µm. Consequently, some taxa could be identified to the species level, while others could only be determined at the genus, family, or order level. Some taxa are either too small or could not be precisely recognized by Ecotaxa for other reasons (e.g., sample quality, image quality during Zooscan scanning) and therefore were not identified to species. Instead, they were grouped at the finest taxonomic level that Ecotaxa could assign, which in some cases is only the order. For example, 65 % of the total copepods were classified as "Calanoida undetermined" for these reasons. Table 2 therefore does not show all recognized taxa within each of the eight categories, but only those that account for at least 1 % of the total abundance within that specific category.

- Problematic trophic categories: many behavioural studies have demonstrated that copepods do not "filter" the food particles as pelagic tunicate instead do. Some copepods create feeding currents that convey water with food particles to the mouth appendages, and *Temora* is one example. *Acartia* and *Centropages* instead have a mixed feeding strategy, they can switch from feeding currents to ambush predation, according to the

type of prey prevailing in the environment. These three genera have been pooled in the "Copepod filter-feeders" category in Table 2, where also *Pleuromamma* is included. But *Pleuromamma* swims very fast, with a motion behavior that does not allow creating feeding currents. Oncaeidae are placed in the category of "Copepods cruise-feeders", but these cyclopoids exhibit a "jerky, hop-and-pause" motion (Hwang & Turner, 1995) as it clearly appears from observing live oncaeids.

→ Given these inaccuracies, trophic groups add little value and risk misinterpretation. Therefore, I recommend to 1) remove section 3.4.3 (trophic groups) and Figure 8, 2) focus the analysis on taxonomic composition, ensuring all major taxa (including *Clausocalanus* and *Ctenocalanus*) are properly represented. If trophic roles are critical to interpret the community distribution patterns, discuss them in the Discussion section, citing behavioral literature to support functional interpretations.

We agree that the trophic categories as originally presented were problematic. Accordingly, we have removed all trophic-related content, including the corresponding column in Table 2, section 3.4.3, and Figure 8. The analysis now focuses on taxonomic composition.

This limitation is mainly due to the fact that most calanoid copepods could not be identified beyond the order level, while they span across all trophic groups. Therefore, the taxonomic resolution was not sufficient to assign trophic categories in a reliable way.

**Additional needs of improvement:**

-The title reflects the content but could be more engaging by briefly highlighting the main finding (e.g., the NBF's role as a boundary rather than an accumulation zone).

We agree and propose the following revised title:

"The North Balearic Front as a zooplankton boundary: fine-scale distribution patterns in late spring."

-In the abstract, "largely unknown" (line 4) should be replaced by "still insufficiently known" for greater precision.

**This change has been made as suggested.**

- Regarding the language, the manuscript requires thorough English editing to improve clarity and flow. The current style is heavy, with redundancies and repetitions. The Results section is overly detailed and should be streamlined for conciseness.

**The manuscript has been revised accordingly**

- In synthesis, this study provides useful data on zooplankton distribution across the NBF but would be significantly improved by a substantial revision focused on 1) refining the taxonomic resolution, 2) removing the trophic classification, 3) tightening the writing.

These points have been revised in the new version of the manuscript.

**Specific comments:**

**Introduction**

1. The current description of the study area (lines 51-66) is too detailed for the Introduction. Please reduce this to just a few lines that introduce the study's aims. The detailed geographical and hydrological information should be moved to a dedicated "Study Area" subsection (2.1) in Materials

and Methods. This subsection should be separate from the sampling strategy and should include a map of the northwestern Mediterranean showing the key hydrological structures and the BioSWOT-Med survey area (clearly framed).

Map suggestion:

**Figure 1.** Maps of the NWMS showing the major oceanographical currents and front (NC: Northern Current, BC: Balearic Current, NBF: North Balearic Front, WMDW: Western Mediterranean Deep Water formation area) of the northern part of the NWMS. After Millot (1987), López García (1994) and Pinardi and Masetti (2000).

- 2. At the end of Introduction, 1) the current questions about zooplankton communities should be preceded by a brief description of the cruise's general interdisciplinary scope; 2) the zooplankton study aims should be presented with clear hypotheses rather than only questions.
- 3. -Improve paragraph flow by moving the first sentence to the end of the first paragraph (better transition to paragraph 2. The second sentence works well as the new opening sentence about frontal zones.
- **4.** Line 35: Avoid repetition suggest: "...concentrate high phytoplankton abundance, supporting elevated zooplankton stocks and metabolism..."
- 5. Line 41: Simplify to "...and their predators..."
- **6.** Line 44: Add DVM abbreviation at first mention: "...in zooplankton Diel Vertical Migration (DVM) have also been observed...
- 7. Line 45-46: Clarify to "...investigating zooplankton distribution at fine scales..."
- **8.** Line 46: Specify what "particles" refers to (e.g., potential prey items?)
- **9.** Line 48: Clarify what "varying widths" describes (height of biomass peaks? frontal features?)
- **10.** Line 67: Briefly present the cruise's interdisciplinary scope before detailing the zooplankton study aims, which should be hypothesis-driven

We thank the reviewer for these detailed suggestions. All points have been adressed: the study area is now in a dedicated subsection, the cruise scope and zooplankton hypotheses are clarified, paragraph flow improved, repetitions reduced, and terms and abbreviations specified.

**Mat & Met:**

1. For better clarity and structure, subsection 2.1 should be divided into: 2.1 Study Area, and 2.2 Sampling Strategy (not "sample strategy").

We agree and have split subsection 2.1 accordingly.

2. The sampling approach needs a clarification: the mention of "drifting stations" (line 89) suggests a Lagrangian sampling strategy. Please clarify this point.

We have clarified the use of drifting stations in the sampling approach.

3. The bathymetric range of the sampled area should be provided.

We have added the bathymetric range of the sampled area.

4. Include details on how the filtered water volume was measured for zooplankton tow

The filtered water volume was not measured with a flowmeter but estimated from the net and towing distance.

5. Lines 76–77: This sentence appears to be a figure caption and should either be removed or rephrased.

This issue has been resolved.

6. Line 78: Explicitly define the acronym SWOT.

The acronym SWOT has now been defined.

7. Line 79: Clarify what is meant by "high spatial resolution" by providing specific values.

We have clarified this, specifying the high spatial resolution as 2 km.

8. Line 84: Revise to: "...physical, chemical, and biological properties."

This revision has been made.

9. Line 85: Specify the range of "fine scale" (e.g., meters, kilometers).

The range of "fine scale" is now specified as kilometers (typically 2 km)

10. Line 94: Replace "sampled" with "recorded" (CTD measures properties, does not "sample" them).

This revision has been made.

11. Table 1: Add a column indicating sonic depths.

If this refers to the bottom depth, the modification has been made (see below)

| Campaign Stage      | Station Name | Water Mass | Date - Time   | Latitude | Longitude | Depth (m) |
|---------------------|--------------|------------|---------------|----------|-----------|-----------|
|                     | a1_D         | A          | 25/04 - 12:38 | 41.240   | 4.553     | >2500     |
|                     | a1_N         | A          | 26/04 - 00:02 | 41.224   | 4.563     | >2500     |
| 1st transect        | f1_D         | Front      | 26/04 - 12:11 | 41.099   | 4.423     | >2500     |
| 1st transect        | f1_N         | Front      | 27/04 - 00:32 | 41.102   | 4.456     | >2500     |
|                     | b1_N         | В          | 28/04 - 00:17 | 40.874   | 4.388     | >2500     |
|                     | b1_D         | В          | 28/04 - 12:28 | 40.884   | 4.389     | >2500     |
| Storm               | m_N          | M          | 02/05 - 00:37 | 39.555   | 4.101     | 1350      |
| 501111              | m_D          | M          | 02/05 - 12:22 | 39.493   | 4.087     | 1500      |
|                     | b2_D         | В          | 04/05 - 12:16 | 40.795   | 4.933     | >2500     |
|                     | b2_N         | В          | 05/05 - 00:13 | 40.849   | 4.936     | >2500     |
| 2nd transect        | f2_D         | Front      | 05/05 - 11:49 | 41.175   | 5.108     | >2500     |
| Ziid ti alisect     | f2_N         | Front      | 06/05 - 00:45 | 41.134   | 5.308     | >2500     |
|                     | a2_N         | A          | 07/05 - 00:13 | 41.412   | 5.24      | >2500     |
|                     | a2_D         | A          | 07/05 - 12:15 | 41.376   | 5.253     | >2500     |
|                     | m2_D         | M          | 10/05 - 11:31 | 39.671   | 3.957     | 1150      |
| Return water mass M | m2_N         | M          | 11/05 - 00:31 | 39.629   | 3.902     | 1200      |
|                     | m2_D'        | M          | 11/05 - 11:53 | 39.603   | 3.885     | 1300      |
| Return water mass B | b3_D         | В          | 12/05 - 12:15 | 40.782   | 5.152     | >2500     |
| Neturn water mass b | b3_N         | В          | 12/05 - 23:58 | 40.746   | 5.112     | >2500     |

Table 1. Station details. In Station Name, 'D' stands for Day and 'N' stands for Night. Depth values are approximate (±50 m) for the station within the water mass M. Depths indicated as ">2500" correspond to stations deeper than 2500 m.

12. Line 105: State the name and location of the shore-based laboratory.

This issue has been resolved.

13. Line 106: Clarify whether "each sample" refers to: separate samples from the 200 μm and 500 μm nets, or a merged sample combining both.

This revision has been made.

14. Line 109: Explain how the approximate number of individuals (~1500) was determined a priori.

This has been clarified.

15. Table 2: See General Comments regarding content. In addition, the trophic group "All" (including "Other organisms") is misleading; I suggest renaming it "Undefined"

This has been taken into account, as Table 2 was revised following the removal of the trophic groups as well as certain taxonomic groups identified by Zooscan.

16. Line 125: Rephrase for clarity. Were the 200 μm and 500 μm samples merged before ZooScan analysis, or were they processed separately and the counts later combined (summed)?

This has been rephrased: "The 200  $\mu$ m and 500  $\mu$ m net samples were processed separately using ZooScan, and their resulting counts were subsequently combined"

17. Subsection 2.5: The method of deriving zooplankton abundance/composition for the layers 100–200 m and 200–400 m by subtracting data from 0–100 m, 0–200 m, and 0–400 m is unconventional. While inevitable due to the sampling design and gears, this approach introduces potential errors (e.g., contamination between layers, negative abundances, as observed here for Eumalacostraca and Cnidaria). Please discuss: the limitations of this method, and how potential biases were addressed (or acknowledged).

We acknowledge that patchiness certainly leads to significant variations in abundances between hauls, as the three net hauls were carried out within two hours. This patchiness is visible in Figure 2, where abundances do not always decrease consistently with depth. In particular, abundances in the 100–200 m layer may be misestimated at certain stations. To provide context, we can compare our data with reference values from Di Carlo (1984), which report relative abundances of approximately 57% in 0–100 m, 27% in 100–200 m, and 16% in 200–400 m.

In our study, the observed mean relative abundances were  $46.2 \pm 18.2\%$  in the 0–100 m layer,  $26.9 \pm 18.5\%$  in the 100–200 m layer, and  $26.8 \pm 15.5\%$  in the 200–400 m layer.

While Di Carlo (1984) did not differentiate between day and night sampling and used different net mesh sizes, this comparison provides useful context.

We also note that abundances in the 0–100 m layer are accurate, and no concerns apply to analyses for this layer; uncertainties remain for the two deeper layers, which cannot be fully resolved.

18. Line 155: Why were eight copepod taxa selected as the "most abundant" rather than another number (e.g., ten)? Clarify: the percentage these eight taxa represent within the total copepod assemblage, and the rationale behind choosing this specific number.

For the analyses, only copepod subgroups with abundances greater than 1% of the total copepods were considered.

**Results:**

- 1. I recommend streamlining the text to improve clarity and flow while preserving key findings (with the exception of subsection 3.4.3, which should be reconsidered -see below).
- 2. Given the methodological concerns raised in my General Comments, I suggest focusing the community composition analysis on taxonomic groups only, removing section 3.4.3 (trophic groups) and the related Figure 8. Discussion of trophic roles, particularly for key groups influencing zooplankton distribution across water masses, should instead be addressed in the Discussion section.

We have removed subsection 3.4.3 and all analyses on trophic groups, as suggested.

3. Additionally, the term "intermediate" (when referring to depth layers) is unnecessary and potentially misleading; it should be removed from both the text and figure captions.

The term "intermediate" has been removed from the text and figure captions.

4. Line 239: Specify that the data refer to the 0–200 m depth layer.

This has been specified.

5. Line 241: From Figure 3, it appears that the 500 μm mesh net also effectively captures Appendicularia and Chaetognatha, not just the listed taxa. Include these groups.

Appendicularia and Chaetognatha have been added as suggested.

6. Line 247: Remove the speculative statement "This intermediate layer likely reflects a transitional zone where DVM results in taxonomic shifts." Such interpretations should be moved to Discussion.

The statement has been removed from the Results section.

- 7. Section 3.3: This section is overly verbose and should be condensed.
  - Section 3.3 has been condensed for clarity, notably by removing the analysis based on biovolume (former Fig. 2b and 2c), in line with the suggestions of the second reviewer.
- 8. Lines 252–256 are redundant. The term "less structured composition" is vague, define what this means. The link to diel vertical migration (DVM) is speculative without direct evidence. Move this discussion to the Discussion section. The details on layers and Hellinger distances would be better organized in a table for clarity.

**These points have been addressed**

9. Line 395: The claim that the cruise occurred during the "the post-bloom period, when phytoplankton biomass levels are already too low to sustain optimal growth of specific zooplankton groups" lacks supporting data. Either provide referenced evidence, or remove the statement unless it can be substantiated by presenting data.

The convection occurred in mid-February 2023 (at Mouillage Lion, see figure below; A. Bosse, pers. comm.), and restratification took place in April, so the cruise occurred after the bloom peak.

Moreover, chlorophyll fluorescence profiles are provided in Supplementary Figure A3, showing a deep chlorophyll maximum at all stations with concentrations around 0.5–0.7 mg/m³, which is low.

**Figures:**

1. Figure 2: The current caption is confusing and needs revision.

The biovolume analyses have been removed following Referee 2's comments.

Clarify if "Total abundance" includes all organisms across the full-size spectrum?

Yes, "Total abundance" includes all organisms across the full-size spectrum. Clarified in the text.

2. The caption reports ESD (Equivalent Spherical Diameter), while the text refers to ECD (Equivalent Circular Diameter) (Lines 110, 128, 132, 133). Ensure consistency.

This has been corrected for consistency between caption and text.

3. Asterisk means "the net could not be analyzed," yet data appear in the histogram. Revise or clarify this discrepancy.

Revised: "The asterisk (\*) indicates that the 200–400 m net at station m\_N could not be analysed."

4. The stations located at the front should be more evidently and immediately identifiable.

Letters indicating water mass have been added to Figure 2 (see above)

5. Figure 3: For easier interpretation, reorganize the histograms so that Copepoda are at the base, followed by Cnidaria, Thaliacea, and other groups.

The histograms have been reorganized as suggested:

Figure 4

6. Figures 5, 9: Add letters (a, b, c) to distinguish the three panels clearly + add depth

Letters and depth information have been added to the panels as suggested.

7. Figure 6: The bars are too small and hard to distinguish. Enlarge or adjust for better readability.

Figure 6 has been redone using the same representation as Figures 3 and 4, as the original boxplot was not correct. Referee 2's comment :

**"Figure 6**: If there are only three samples per station (corresponding to depth layers), boxplots may not be appropriate (a boxplot summarises a data distribution with 5 values; if you have only 3, this does not make sense). Consider an alternative method of data representation if that is indeed the case.

Figure 6 has been replaced by a new version (see below). The boxplot in the first version used data from both nets (distinguishing 200 µm and 500 µm mesh, as in Figure 3). Since all results are now presented for the merged nets, the figure was replaced by one averaging the three depth layers and ordering stations chronologically, consistent with Figure 4. The previous figure was clearly not valid based on the merged data, and we thank the reviewer for pointing this out."

8. Figure 8: It should be removed (see my comments above on the trophic traits).

Figure 8 has been removed, as suggested.

**Discussion:**

1. The Discussion is well-developed but could be strengthened with more detailed insights on the species-/genus-level distribution patterns of zooplankton, which would better elucidate adaptations to different water masses. Some structural improvements are necessary.

We have revised the Discussion following your comments, as explained below. However, for the species-/genus-level analyses are not really possible, as explained previously in relation to the Ecotaxa classification.

2. Currently, there is unnecessary mixing of result interpretation and comparisons with previous studies (e.g., second paragraph of Section 4.1). These should be separated for clarity.

We have revised the section to separate result interpretation from comparisons with previous studies.

3. A summary table comparing zooplankton abundance/biomass with prior studies in the region would be more effective than textual descriptions.

A summary table has been added (see below) as suggested.

| Campaign    | Season            | Region                       | Location                              | Concentration (ind/m³) | Biomass
(mg DW/m³) | Depth range (m) |
|-------------|-------------------|------------------------------|---------------------------------------|------------------------|-----------------------|-----------------|
| DEWEX 2013  | Winter (February) | DCZ (A)                      | Near LION Station (42°04' N, 4°38' E) | 200                    | 5                     | 0-250           |
|             |                   | DCZ Periphery / Balearic (B) | North of Menorca Island               | 650                    | 10                    | 0-250           |
|             | Spring (April)    | DCZ (A)                      | Near LION Station                     | 4400                   | 100                   | 0-250           |
|             |                   | DCZ Periphery / Balearic (B) | North of Menorca Island               | 2000                   | 30                    | 0-250           |
| BioSWOT-Med | Late Spring (May) | Water mass A (Transect 1)    |                                       | $1848 \pm 133$         | $29 \pm 4$            | 0-200           |
|             |                   | Water mass B (Transect 1)    | see Table 1                           | 881 ± 212              | 8 ± 3                 | 0-200           |
|             |                   | Front F (Transect 1)         |                                       | 615 ± 44               | $9\pm2$               | 0-200           |
|             |                   | Water mass A (Transect 2)    |                                       | 745 ± 27               | $7\pm2$               | 0-200           |
|             |                   | Water mass B (Transect 2)    | see Table 1                           | $333 \pm 9$            | $14 \pm 3$            | 0-200           |
|             |                   | Front F (Transect 2)         |                                       | 983 ± 155              | 6 ± 1                 | 0-200           |

4. The discussion on zooplankton biomass drivers at fronts (4.2) and the front's role as a mixing zone vs. distinct community boundary (4.3) should be merged and condensed to avoid redundancy.

We acknowledge the reviewer's concern about redundancy. While sections 4.2 and 4.3 have not been merged, both have undergone a substantial reorganization and rewriting in order to improve clarity and reduce overlaps.

**Further Comments:**

5. Line 406: What explains the contrasting responses of Cnidaria/Foraminifera (positively influenced by the front) vs. Thaliacea (negatively affected)? An attempt of explanation is needed.

Our potential hypotheses are as follows: In our observations, Cnidaria and Foraminifera mainly consisted of small organisms (e.g., Cnidaria were mostly ephyrae) with limited swimming capacity and a carnivorous trophic behavior on small prey. These organisms, and their prey, are likely favored by the accumulation of resources at the front. In contrast, Thaliacea were mostly individuals from salp chains, which display large-amplitude diel vertical migrations. These organisms may actively avoid some of the physical (e.g., turbulence) and trophic (e.g., high particle load) conditions typically observed at fronts.

6. Lines 414–415: The overly generic statement "These patterns likely result from interactions between species-specific behaviors and frontal dynamics" should be **r**ephrased with more precise reasoning (e.g., citing known behavioral or hydrographic drivers).

This statement has been rephrased:

'In other frontal studies, some taxa have been found more abundant than in adjacent waters (Molinero et al., 2008). Gastauer and Ohman (2024) similarly reported front-related increases in appendicularians, copepods, and rhizarians, underscoring that zooplankton community composition is shaped by species-specific responses. Biomass peaks also depend strongly on the taxa considered (Mangolte et al., 2023). However, in our analyses, we never focus on a single taxon, but rather on groups of organisms (Table 2) or on the whole sampled mesozooplankton.'

7. Lines 423–424: Is there a hypothesis for why certain taxa (Magelonidae, cyphonautes, echinoderm larvae, radiolarians, Heteronemertea) were absent at the front? If speculative, frame it as a question for future research.

The apparent absence of some taxa (Magelonidae, cyphonautes, echinoderm larvae, radiolarians, Heteronemertea) at the front most likely reflects the limitations of semi-automatic identification in EcoTaxa rather than a true ecological pattern. Moreover, these taxa were extremely rare in the other water masses as well, with only a few individuals observed. This sentence has been then removed.

8. Section 4.5: The title "Storm Impact?" should be assertive (e.g., "Potential Storm Effects")

In the revised version, the section originally entitled "Storm Impact?" has been removed. Instead, we created a broader section entitled "Confounding factors affecting zooplankton structure", in which the potential influence of the storm as well as diel vertical migration are now addressed.

9. Line 461: Provide a reference for the chl a-fluorescence glider data

A reference has been added: "A. Bosse, pers. comm."

**General technical notes:**

1. In Methods and Results, all verbs should be in the past tense, while some are now erroneously in the present tense.

This has been corrected

2. In Methods and results, some taxonomic categories are given in Latin, while others are in English. Ensure uniformity throughout the manuscript (text, tables, figures).

This has been corrected

3. Maintain consistency throughout the manuscript, always writing the acronyms (which should be made explicit only at the first citation).

This has been corrected

---

## Author Comment (AC2)

We would like to thank the reviewer for the thorough and constructive comments. These suggestions have been carefully considered and have helped us to improve the manuscript. Below, we provide a detailed response to each comment.

**Main concern**

The clarity and impact of the findings are hindered by a presentation that is at times diffuse and by the inclusion of numerous analyses whose relevance is not always made explicit.

The manuscript would benefit from a more focused and structured approach, with a clear emphasis on its main scientific question: the evaluation of zooplankton community responses to a frontal system (and the conclusion that the community is not just a mix of those of the surrounding water masses).

Its length could probably easily be reduced by \( \frac{1}{3} \), possibly \( \frac{1}{2} \).

The inclusion of data from the "M" stations introduces complexity that does not clearly support the core narrative. These stations appear to add variability that confounds rather than clarifies the analysis of the frontal signal. In particular, the inclusion of M stations significantly alters multivariate patterns such as those visible in the PCAs in Figure 5 and following, making it more difficult to discern the contrasts between the A, B, and F stations, which are those associated with the frontal gradient. I would strongly recommend excluding the M stations entirely from the analyses and focusing the manuscript on the transects most relevant to the front.

We agree with the reviewer that including the M stations in certain analyses (mainly PCA) can obscure the frontal signal. However, the results from the M stations, relatively to the other stations, are useful for other outcomes of the BioSWOT-Med campaign (e.g., Zooglider, fluxes). Consequently, we propose to retain the results from the M stations for concentration and taxonomic distribution (Figures 2, 4, and 6) in the main manuscript. To follow reviewer comment, the M stations will not be considered in the analyses presented in Section 3.4 of the main manuscript (PCAs in Figures 5, 7, and 9) focused on the frontal signal.

Moreover, we propose to present the positions of the M stations in the supplementary materials (PCA, etc.), treating these points as supplementary individuals.

See PCAs below:

Same PCA as Figure 5 in the first version of the manuscript, but without the M stations.

Similarly, we have the same PCA as Figure 7 in the first version of the manuscript, but without the M stations.

Same PCA as Figure 9 in the first version of the manuscript, but without the M stations.

For the Supplementary Materials, the two previous PCAs are shown again, but with the M stations projected as supplementary individuals:

Moreover, while the breadth of analyses presented is commendable, their number may overwhelm the reader and dilute the central message. A clearer selection of the most pertinent analyses to highlight your main message (e.g., PCA on concentration or biovolume, barplots, NBSS) would help maintain the reader's focus.

While PCA on concentration or biovolume alone does not yield meaningful projections, we have revised the manuscript to reduce the number of analyses (removed Fig. 2b, 2c) and retain only the most pertinent ones, ensuring a clearer focus on the main message.

Other analyses—such as those related to diel vertical migration or the storm—could be treated as potential confounding factors. These could then be addressed more briefly in the discussion with some plots presented in supplementary materials if necessary. In addition, the manuscript attributes several observed differences between cruise legs to the passage of a storm. However, the current dataset does not make it possible to unambiguously separate storm effects from general temporal or spatial variability. This uncertainty should be acknowledged more explicitly in the discussion, and interpretations emphasising the storm as a dominant factor should be tempered accordingly.

The Discussion has been revised following these general comments and the more specific suggestions detailed below.

**Major comments**

Line 113: The authors limit the analysis to eight categories derived from the ZooScan. It is unclear why only these eight were used, given that the system allows for much finer taxonomic resolution. These categories were likely aggregates of finer taxonomic groups. Why didn't you use the finer level data? Indeed, as acknowledged in lines 423–425, the inclusion of rarer taxa enhanced the ability to distinguish between water masses, indicating that finer-scale groupings may be more informative for detecting ecological responses to fronts. Coarse groupings such as "copepods" may be too ubiquitous to reveal significant patterns.

We used eight broad categories because Ecotaxa assigns organisms at varying taxonomic levels (species, family, order), and this grouping ensures consistency.

Moreover, the semi-automatic taxon recognition process was performed on Zooscan images with a pixel size of 10.3 µm. Consequently, some taxa could be identified to the species level, while others could only be determined at the genus, family, or order level. Some taxa are either too small or could not be precisely recognized by Ecotaxa for other reasons (e.g., sample quality, image quality during Zooscan scanning) and therefore were not identified to species. Instead, they were grouped at the finest taxonomic level that Ecotaxa could assign, which in some cases is only the order. For example, 65 % of the total copepods were classified as "Calanoida undetermined" for these reasons.

Following your comments, we performed PCAs (see below) subdividing the copepods into the seven categories defined by EcoTaxa for the most abundant copepods (seven categories because only those with >1% of the total copepod abundance were retained): Undetermined Calanoida, *Centropages* spp., *Corycaeidae*, *Euchaeta*, *Oithona*, *Oncaeidae*, and *Pleuromamma* spp.

Lines ~145: The vertical distribution estimation based on two net tows at different depths, with subtraction, is methodologically weak compared to dedicated stratified sampling using e.g. a multinet. Variability in the upper layer between tows could significantly affect results. Ideally, replicate shallow tows should be presented to estimate intra-station variability and compare it with the inter-station variance. If this is not possible, a discussion of this methodological limitation should be included, supported where possible by literature.

We acknowledge that patchiness certainly leads to significant variations in concentrations between hauls, as the three net hauls were carried out within two hours. This patchiness is visible in Figure 2, where concentrations do not always decrease consistently with depth. In particular, concentrations in the 100–200 m layer may be misestimated at certain stations. To provide context, we can compare our data with reference values from Di Carlo (1984), which report concentrations of approximately 57% in 0–100 m, 27% in 100–200 m, and 16% in 200–400 m with respect to the 0-400 m layer.

In our study, the observed mean concentrations were  $46.2 \pm 18.2\%$  in the 0–100 m layer,  $26.9 \pm 18.5\%$  in the 100–200 m layer, and  $26.8 \pm 15.5\%$  in the 200–400 m layer with respect to the 0-400 m layer.

These numbers show that, on average, the concentrations follow a reasonable pattern but with significant departures to the average situation.

As Di Carlo (1984) did not differentiate between day and night sampling and used different net mesh sizes, this comparison is only an indication.

We add in discussion that concentrations in the 0–100 m layer are accurate, while uncertainties remain for the two deeper layers, which cannot be fully resolved.

**Line 173**: The use of the Hellinger transformation shifts the analytical focus to relative composition rather than absolute concentrations. This is not inherently "preferable" to the use of Euclidean distance on raw data, but represents a different analytical approach. This choice should be justified explicitly, as it influences the interpretation of the results.

The Hellinger transformation was used to focus on relative abundances, which allows all data to be analyzed together. Using absolute abundances would mainly discriminate between the first and second transects and would not reveal a stable gradient between water masses. This choice is now explicitly justified in the manuscript.

Lines ~215: Since the fT stations are constructed as linear combinations of stations A and B, their positions in PCA space should likewise fall between the positions of those stations. Due to the Hellinger transformation, this may not be a straight line but a curved path. Still, the optimal mixing between a and b should be computable exactly (i.e. without iteration) from the PCA space. In any case, recalculating the PCA with each added station alters the structure of the PCA space (even if this is likely small here) and this hinders comparisons. It would be more appropriate to construct a PCA using A, B, and F stations, and then project the fT stations as supplementary points.

Thank you for this remark and sorry for our unclear explanation. Our method does exactly what you mentioned: fT stations are only projected as supplementary points on the initial PCA axes. They do not influence the axes or the positions of the actual stations. No PCA recalculation is performed; the loop is used only to find the minimum of the cumulative f–fT distances. This procedure is now explained more clearly in the manuscript.

**Section 3.3**: While this section demonstrates the presence of diel vertical migration, it does not quantify its importance relative to other sources of variability. A multivariate analysis such as PCA including all stations could help illustrate whether day and night samples from the same station are more similar to one another than to samples from other stations (and you have it... so cite it here).

In the global PCA (Figure 5), differences between water masses dominate axis 1, while diel vertical migration (DVM) is mainly represented on axis 2. This shows that DVM is a secondary source of variation compared to spatial differences between stations. Day and night samples from the same station (x1-D and x1-N) are generally close, and there is no grouping of day or night samples across different stations.

We also explored a PCA based on biovolume or abundance alone, as previously suggested, but this did not yield meaningful results, since the projections were too strongly clustered. Overall, the global PCA allows us to better quantify the variation induced by DVM, confirming that it is not the primary structuring factor.

**Section 3.4.3**: This section appears to replicate earlier PCAs using trophic groups rather than taxonomic ones. It is not clear what new insight this re-analysis is intended to provide. If there is a hypothesis suggesting that trophic groups respond differently to frontal structures, it should be clearly stated. Otherwise, the patterns described may simply reflect underlying taxonomic distributions.

This concern was also raised by the other reviewer. As a result, the trophic group analyses, including Section 3.4.3 and Figure 8, have been removed from the manuscript.

Lines 479 and 484: The discussion mentions the value of high-resolution observations and autonomous platforms. If I am not mistaken, a Zooglider was deployed during the BioSWOT campaign. It would be extremely valuable to discuss these data alongside the net samples. The integration of these two datasets could significantly enhance the interpretation of the observed patterns compared to studying them in isolation. I was actually expecting to read about this when I accepted the review and was disappointed to see only the net data.

Unfortunately, due to logistic contrains in France, Zooglider was deployed south of Majorca by spanish colleagues, and thus could not sample the front as expected. Most of the glider transects are south of Majorca and Menorca, except some transects near station B3. Therefore, M stations and B3 will be useful for comparison with these data.

The Zooglider data will be analysed in a separate study.

**Detailed comments**

• The manuscript contains inconsistencies in citation formatting. References should follow the format (Author Year) rather than (Author (Year)).

All references will be formatted according to the (Author Year) style.

• Line 36: The assertion that fronts concentrate plankton should be moderated, as this is not always the case. Indeed, you already provide a more nuanced statement just a few lines above.

The statement has been moderated accordingly.

• **Figure 1**: It would be helpful to include a contextual map showing general ocean circulation as introduced in the background. The ship's trajectory should also be overlaid. Each subpanel should show only the relevant stations for its respective transect.

A map illustrating general ocean circulation has been prepared (see below). Overlaying the ship's trajectory on Figure 1 would make it too busy, but it is available in the campaign report: <a href="https://doi.org/10.13155/100060">https://doi.org/10.13155/100060</a> (Figure 7, page 13).

See below the map showing general ocean circulation of the NWMS:

**Figure 1.** Maps of the NWMS showing the major oceanographical currents and front (NC: Northern Current, BC: Balearic Current, NBF: North Balearic Front, WMDW: Western Mediterranean Deep Water formation area) of the northern part of the NWMS. After Millot (1987), López García (1994) and Pinardi and Masetti (2000).

• Line ~80: A clearer description of the software tools used would be beneficial. Their specific functions and utility during the cruise should be outlined.

All SPASSO software used is described in detail in Rousselet et al. (2025).

• Lines 86–88: This material reiterates content already provided in the introduction. It would be advisable to consolidate this information in one section and refer to Figure 1 directly from the introduction.

The information regarding the description of the hydrography of the area has been greatly simplified in the Introduction section and now forms part of a new subsection of the Materials and Methods, *Study area*. The repetition about zones AB and F at line 86 has now been incorporated into this subsection.

• Line 89: If the station drift over 24 hours is not negligible relative to the map scale, this should be depicted as a trajectory rather than as discrete points.

There was a misunderstanding: the map already shows the drift between day and night sampling locations for each station. Only station F2 experienced significant drift.

• Line 90: Two "f2" stations are presented: f2\_D and f2\_N... but the day/night distinction has not been made explicit yet.

The day/night distinction for F2\_D and F2\_N is made explicit in Table 1, cited just before discussing these stations.

• Lines 95–96: The notation for water masses and stations should be presented earlier to guide the reader from the outset.

Notation for water masses and stations are already presented earlier in the first paragraph of the Materials and Methods and are shown in Figure 1

• Table 1: The column listing station names does not provide essential information and could be removed.

The station name column has been removed (see below)

| Campaign Stage      | Station Name | Water Mass | Date - Time   | Latitude | Longitude | Depth (m) |
|---------------------|--------------|------------|---------------|----------|-----------|-----------|
| 1st transect        | a1_D         | A          | 25/04 - 12:38 | 41.240   | 4.553     | >2500     |
|                     | a1_N         | A          | 26/04 - 00:02 | 41.224   | 4.563     | >2500     |
|                     | f1_D         | Front      | 26/04 - 12:11 | 41.099   | 4.423     | >2500     |
|                     | f1_N         | Front      | 27/04 - 00:32 | 41.102   | 4.456     | >2500     |
|                     | b1_N         | В          | 28/04 - 00:17 | 40.874   | 4.388     | >2500     |
|                     | b1_D         | В          | 28/04 - 12:28 | 40.884   | 4.389     | >2500     |
| Storm               | m_N          | M          | 02/05 - 00:37 | 39.555   | 4.101     | 1350      |
|                     | m_D          | M          | 02/05 - 12:22 | 39.493   | 4.087     | 1500      |
| 2nd transect        | b2_D         | В          | 04/05 - 12:16 | 40.795   | 4.933     | >2500     |
|                     | b2_N         | В          | 05/05 - 00:13 | 40.849   | 4.936     | >2500     |
|                     | f2_D         | Front      | 05/05 - 11:49 | 41.175   | 5.108     | >2500     |
|                     | f2_N         | Front      | 06/05 - 00:45 | 41.134   | 5.308     | >2500     |
|                     | a2_N         | A          | 07/05 - 00:13 | 41.412   | 5.24      | >2500     |
|                     | a2_D         | A          | 07/05 - 12:15 | 41.376   | 5.253     | >2500     |
| Return water mass M | m2_D         | M          | 10/05 - 11:31 | 39.671   | 3.957     | 1150      |
|                     | m2_N         | М          | 11/05 - 00:31 | 39.629   | 3.902     | 1200      |
|                     | m2_D'        | M          | 11/05 - 11:53 | 39.603   | 3.885     | 1300      |
| Return water mass B | b3_D         | В          | 12/05 - 12:15 | 40.782   | 5.152     | >2500     |
|                     | b3_N         | В          | 12/05 - 23:58 | 40.746   | 5.112     | >2500     |

• Line 115: The term "abundance" is used, but the actual metric is concentration (ind/m³). This should be corrected throughout the manuscript.

All instances of "abundance" have been replaced with "concentration" throughout the manuscript.

• Line 152: It is not clear what the "groups" are at this point. Overall, in the methods section, I would advise to always start by explaining the ecological purpose of the analyses and only then, describe (which tests, which hypotheses, etc.) you will carry them out. Currently, the rationale of the analyses is often not clear.

The reference to trophic groups has been removed, along with the associated figure, as the trophic analyses have been deleted.

• Line 152: Normality should be assessed on residuals rather than raw data. If normality assumptions were not met, alternative methods should be justified. Indicate whether data transformations were applied to reach normality (it's likely that a transformation was used).

Normality was assessed on residuals; this is now clarified in the manuscript. And normality of residuals was confirmed without the need for any additional data transformation.

• Line 155: The aim of the copepod subgroup test should be more explicitly explained.

The aim of the copepod subgroup test has been clarified in the manuscript; it was conducted to investigate diel vertical migration (DVM) patterns within the copepod community.

• Line 163: The x-axis of the NBSS should be in units of mm3.

We follow the approach of Platt and Denman (1977), as in de Souza et al. (2020), for the NBSS axes; therefore, units are consistent with these references.

• Line 165: The y-axis should be expressed in mm³/m³/mm³; the denominator represents Δvolume in mm³.

The y-axis unit was a typographical error: it was written as "Δvolume.mm-3)" but should be "Δvolume (mm3)"; the values are correctly expressed in m-3.

• Line 167: If the ellipsoid volume approximation is deemed superior, as you state at lines 135-136, explain why the spherical approximation is still used in this section.

The spherical approximation is retained in this section because it is used only to compute the size spectrum from ECD. The ECD is only used for combining the two mesh sizes (200 and 500  $\mu$ m). While the ellipsoid approximation may be more accurate, each biovolume can correspond to multiple combinations of length and width, so the spherical approximation provides a consistent, comparable metric.

• Line ~170: Again, the manuscript should provide the rationale ("why") for each analysis before presenting the methodology ("how").

The rationale for each analysis is now clarified before the methods

• Line 172: Clarify that observed noise in the NBSS at large sizes is due to the rarity of large individuals rather than size per se.

This point has been clarified.

• Line 178: I would suggest replacing the shorthand notation (y1+) with an explicit sum sign. Also, indicate that concentrations—not frequencies—are being summed, and define all variables (y\_ij are not explicitly defined).

This has been clarified: concentrations are indicated, and all variables are defined.

• Line 181: Please explain what is meant by "asymmetric" in this context.

The term "asymmetric" has been clarified:

If two sites both have zero abundance for a species (a double zero), that absence does not contribute to making them more similar. In contrast, with Euclidean distance, double zeros do contribute, which can artificially inflate similarity. Thus, the treatment of presences and absences is not symmetric: presences matter, joint absences don't.

• Line 185: Specify whether normality tests were performed before or after the Hellinger transformation. Note that PCA does not absolutely require normal data but is appropriate only with approximately normal input, so an actual normality test may be excessive. Also, please clarify the purpose of testing correlations between variables (since this seems to me that assessing correlations is what the PCA does already).

Before performing PCA, the Hellinger-transformed data were checked for normality using the Shapiro-Wilk test., and correlations between taxonomic groups were examined to ensure sufficient linear structure for PCA.

• Line 196: Provide details on how "dispersion" was calculated.

Details on how "dispersion" was calculated have been added:

"Dispersion was calculated as the sum of squared Euclidean distances of individuals to their group centroid (intra-group dispersion). Inter-group dispersion was defined as the sum of

squared distances between group centroids and the global centroid, weighted by group size. These measures were used to compute the pseudo-F statistic."

• Line 199: Was the significance of the pseudo-F statistic tested? If so, specify the method.

We did not test the significance of the pseudo-F statistic. It was used here only as a descriptive measure, to give an idea of how different (or not) the water masses are from each other, rather than as a formal test

• Line 202: The notation "fT" is potentially ambiguous. A clearer notation such as f{t}\_D, where {t} is a subscript and indicates theoretical interpolation, would help avoid confusion.

The notation "fT" has been clarified and is now written as f{t}\_D

• Figure 2: Indicate in the axis title that (b) refers to the biovolume of small organisms.

The figures 2.b and 2.c have been deleted, as explained above.

• Figures 3 and 4: Consider using a more refined and perceptually balanced colour palette, such as those offered by Tableau or ColorBrewer.

We have revised Figures 3 and 4 (see below) by applying the ColorBrewer 'Set2' palette to improve perceptual balance and readability:

Figure 3

Figure 4

• Line 252: Why is biovolume analysed here but not in the previous section? Indeed, biovolume provides a valid view of the taxonomic composition. I am not asking for an additional analysis (there are many already); rather I would recommend choosing an angle of analysis, justifying it and sticking by it.

We have removed the biovolume analysis, including Figures 2b and 2c, because biovolume is not suitable for large organisms (e.g., salps, cnidarians, and eumalacostracans) due to high variability and sampling limitations.

• Also, the claimed similarity between different depth layers should be demonstrated using multivariate analyses (e.g., PCA based on Hellinger distances), which would better capture the structure of the data.

We note the suggestion, but additional multivariate analyses were not performed, as we consider them unnecessary for the current focus of the study.

 Line 283: The PERMANOVA test should be described in the methods section. Clarify which factors were tested.

**PERMANOVA and tested factors are now described in Methods**

• Lines 293–294: "This indicates...dynamics of the water masses": I am not sure I understand what you mean. It is unclear why variation in the proportion of group A is interpreted as evidence for vertical migration. Could this not be attributed to bathymetric differences between the regions of A and B water masses for example?

This statement has been clarified and nuanced; but bathymetric differences between A, B, and F water masses do not explain the observed patterns.

• Line 299: The reconstructions of fT station values are assessed on relative concentrations only, within the PCA framework (since the Hellinger transformation was performed). You should not state that you reconstruct "absolute" concentrations.

**The text has been clarified**

• Lines 300 and 303: Specify what the relative increases or decreases are in reference to.

**Noted and addressed**

- Line 312: Avoid abbreviations such as Cop\_CCF or Cni unless defined. Using full names would not be much longer but would be clearer.
  - 3.4.3 has been removed as explained above
- Line 314: Specify what is meant by "non-carnivorous" (e.g., non-carnivorous copepods?).
  - 3.4.3 has been removed as explained above
- Figure 6: If there are only three samples per station (corresponding to depth layers), boxplots may not be appropriate (a boxplot summarises a data distribution with 5 values; if you have only 3, this does not make sense). Consider an alternative method of data representation if that is indeed the case.

Figure 6 has been replaced by a new version (see below). The boxplot in the first version used data from both nets (distinguishing 200  $\mu$ m and 500  $\mu$ m mesh, as in Figure 3). Since all results are now presented for the merged nets, the figure was replaced by one averaging the three depth layers and ordering stations chronologically, consistent with Figure 4. The previous figure was clearly not valid based on the merged data, and we thank the reviewer for pointing this out.

• Figure 9: Much of the description of these results are in terms of shifts between depth layers or between regions, but these are difficult to see since the depth layers are in different subplots. A single PCA plot with region and depth encoded by colour or symbol would facilitate interpretation.

We agree that this could be a useful approach. However, encoding both region and depth in a single PCA plot would result in a cluttered and unreadable figure, particularly in the center of the PCA, and is therefore not feasible.

• Line 337: Rather than referencing previous literature, the nutrient-rich nature of water mass A should be demonstrated using nutrients data collected during the cruise, if possible (I image that basic oceanography variables were collected).

Here is the revised first paragraph of 4.1 below.

The spatial differences between water mass A and B in late spring can be linked to the regional hydrological and ecosystem functioning of the NWMS in the post-bloom period (D'Ortenzio and Ribera d'Alcalà (2009)). Water mass A has its origin in the Liguro-Provencal area (NWMS), characterized by intense convection and mixing (Barral et al. (2021)), high nutrient concentrations (Severin et al. (2017)) and more productivity (Mayot et al. (2017); Hunt et al. (2017)) with the formation of a deep chlorophyll maximum around 50 m (Fig. S3; Lavigne et al. (2015); Doglioli et al. (2024)). Water mass B in the southern part of the NBF comes from the epipelagic waters of the Algerian basin, which are warmer and fresher than waters from the NWMS, with virtually permanent stratification and a DCM deeper than 50 m (Fig. S3; Lavigne et al. (2015)). In the transitional post-bloom period (April-May) encountered during the BioSWOT-Med cruise, water mass A was nutrient-richer than water mass B with mean nitrate (phosphate) concentrations in the euphotic layer ranging 0.64-1.27 (0.003-0.144) µM in A compared to 0.04-0.44 (below detection limit-0.003) µM at B. Those contrasts also appeared at 500 m depth, nitrate (phosphate) concentrations ranging 8.38-9.43 (0.34-0.40) µM in A compared to 7.49-8.89 (0.26-0.36) µM in B (Joël et al., 2025, submitted). Water mass A shows higher zooplankton stocks strongly dominated by copepods and larger forms whereas in water mass B, and community structure is dominated by small sizes and slightly more diversified within the non-copepod organisms (Fig. 4, 9), consistent with Fernández de Puelles et al. (2004).

• Line 369: If prior studies in the NWMS do not address the NBF specifically, this literature review may be condensed.

Noted, the review has been condensed accordingly.

• Lines 370–379: These results from the literature are not clearly linked to your findings. Consider moving this paragraph later, where the discussion is more integrative.

This revision has been made.

• Line 395: Fronts may actually have their strongest effect when nutrients are limiting, such as during the normally post-bloom period of the year, when the cruise occurred. Indeed, they can then enhance nutrient availability and prolong productivity later in the season

**Added**

• Line 408: Clarify what is meant by "higher taxa."

We clarified that 'higher taxa' refers here to the broader taxonomic categories we defined earlier (Table 2)

Line 411: "highlighting the importance of considering individual taxonomy groups rather than just
overall abundance patterns when analysing community dynamics": this claim that taxonomy
matters for community analysis is self-evident: community dynamics is the dynamics of various
species, so, of course, it cannot be assessed with only the overall concentration. Consider
removing or rephrasing.

You are right indeed, it's tautological. This sentence has been removed

• Line 429: The observation that zooplankton differences are stronger at 100–200 m despite the fact that the front is stronger in 0-100m is intriguing and warrants further discussion.

We already mention in the manuscript that the 100–200 m layer likely acts as a transitional zone in the context of DVM, which explains the stronger differences observed there despite the surface front being more pronounced.

• Lines 431–432: "emphasises the stronger influence of hydrology and biological productivity at the surface": and of the front! The fact that the two water masses that meet at the front have a different history is also a good explanation for this observation.

Yes, indeed, this is now clarified in the manuscript

• Line ~440: Diel vertical migration should be introduced early in the results as a potential confounding factor. Explain how this was controlled for/avoided (e.g., comparing only daytime samples) so that you can safely go on with the analyses despite this confounding factor.

Yes, indeed, the analyses concerning DVM have been reshaped, which addresses this comment in particular.

• Line 461: Provide specific details regarding what you observed on the chlorophyll a fluorescence profiles.

We observe no dilution of the DCM after storm. Added in the manuscript.

• **Table 3**: Ensure that this table is referenced appropriately in the text. Note that *Centropages typicus* should be italicised and use lowercase for the species epithet.

This issue has been resolved, see new table below:

| Type of analysis         | Depth   | Taxonomic group / species | ANOVA p-value | p-value
A1st vs A2nd | p-value
B1st vs B2nd | p-value
F1st vs F2nd | p-value
M1st vs M2nd |
|--------------------------|---------|---------------------------|---------------|-------------------------|-------------------------|-------------------------|-------------------------|
| ANOVA Depth              | 0-100 m | Appendicularians          | 0.124         |                         |                         |                         |                         |
|                          |         | Chaetognatha              | 0.039 *       | <0.001 ***              | 0.0659                  | <0.001 ***              | 0.108                   |
|                          |         | Cnidaria                  | <0.001 ***    | <0.001 ***              | <0.001 ***              | 0.418                   | 0.765                   |
|                          |         | Copepoda                  | <0.001 ***    | 0.189                   | <0.001 ***              | <0.001 ***              | 0.617                   |
|                          |         | Eumalacostraca            | 0.534         |                         |                         |                         |                         |
|                          |         | Foraminifera              | 0.429         |                         |                         |                         |                         |
|                          |         | Other_organisms           | 0.375         |                         |                         |                         |                         |
|                          |         | Thaliacea                 | 0.929         |                         |                         |                         |                         |
| ANOVA Copepod
species | 0-100 m | Calanoida                 | 0.255         |                         |                         |                         |                         |
|                          |         | Centropages spp.          | 0.014 *       | 0.002 **                | 0.104                   | < 0.001 ***             | 1                       |
|                          |         | Corycaeidae spp.          | 0.0104 *      | < 0.001 ***             | 0.797                   | < 0.001 ***             | 0.992                   |
|                          |         | Euchaeta                  | 0.581         |                         |                         |                         |                         |
|                          |         | Oithona                   | 0.0231 *      | 0.448                   | 0.0197 *                | 0.923                   | 0.876                   |
|                          |         | Oncaeidae                 | 0.015 *       | < 0.001 ***             | 0.031*                  | 0.025 *                 | 0.87                    |
|                          |         | Pleuromamma spp.          | 0.928         |                         |                         |                         |                         |

• Line 484: Satellite data are mentioned but not utilised. If available, these should be incorporated into the analysis or explicitly discussed.

The use of satellite data is outside the scope of this paper, but it will be mentioned and presented in future works from the BioSWOT-Med cruise.

---

## Author Response (AR2)

**Authors' response**

**We would like to thank the reviewer for the final review. These suggestions have been carefully considered and have helped us improve the manuscript. Below, we provide a detailed response to each comment.**

**Minor points that were overlooked:**

-In the Results section, the term "intermediate" should be removed from the captions of Figures 3 and 5, where it refers to the three depth layers.

The term "intermediate" has been removed and replaced with "reconstructed".

-The 2.2 section title should be "Sampling strategy" and not "Sample strategy".

The Section 2.2 title has been changed accordingly.

**Technical notes**

-In Introduction (L48-54), the fourth paragraph should be restructured for better logic. I suggest moving the second sentence (L50-52) to the beginning. This would present the Northwestern Mediterranean Sea (NWMS) before introducing the BioSWOT-Med cruise, creating a more natural flow and a better link to the following paragraph where the cruise is detailed. Also, the acronyms NWMS and Northern Balearic Front (NBF) should be written in full upon first mention here.

This paragraph has been modified as suggested.

-There is inconsistency in the font style for station names, which appear in both italics and regular font throughout the text, figures, and tables. This should be standardized.

The font style for station names is now standardized.

-Zooplankton concentration is reported as "number of individuals m$^{-3}$" on line 124 and in Table 3, but as "ind / m$^2$" in Figure 3. The units must be consistent.

The use of different units in Figure 3 and Table 3 is intentional and reflects the different objectives of these representations.

Figure 3 displays zooplankton abundance integrated over depth (individuals m$^{-2}$) for each sampled layer. This choice was made because the vertical layers have different thicknesses (100 m, 100 m, and 200 m). Expressing the data in individuals m$^{-3}$ in a stacked bar plot would be misleading, as the height of each segment would then represent density rather than the actual number of individuals contained within each layer, resulting in segments that are not proportional to true abundance.

For these reasons, we consider that using depth-integrated abundance (ind m$^{-2}$) in Figure 3 is the most appropriate and least misleading representation of the data, and we therefore do not modify this figure. Conversion to volumetric concentration (ind m$^{-3}$) is straightforward, as it simply requires dividing the depth-integrated values by the thickness of the corresponding layer (100 m or 200 m).

In contrast, Table 3 reports zooplankton concentration in individuals m$^{-3}$ to allow direct comparison with previously published data from the DEWEX study, which are expressed in individuals m$^{-3}$.

-Figure 3: The number "600000" on the y-axis is out of scale and should be removed. The letters A, B, and F (denoting water masses) should be explained directly in the graph or its caption. Each figure must be self-explanatory.

Figure 3 and its caption have been changed.

-Section 3.4.2: I suggest changing the title from "Comparison of the front community composition with adjacent waters" to "Comparison of the community composition between the front and adjacent waters" for better phrasing.

The 3.4.2 section title is now "Comparison of the community composition between the front and adjacent waters".

-Line 387: The genus names "Pleuromamma and Euchaeta" must be written in italics.

The change has been made.

-Table 4: The title of the third column should be simplified to "Taxa" since different taxonomic levels are listed; "Appendicularia" instead of "Appendicularians"; "Euchaeta" must be in italics; "spp." applies to a genus and should be removed from the family "Corycaeidae".

Table 4 has been modified accordingly.